# Failure by Interference: Language Models Make Balanced Parentheses Errors When Faulty Mechanisms *Overshadow* Sound Ones

**Daking Rai**
George Mason University
drai2@gmu.edu

**Samuel Miller**
George Mason University
smille20@gmu.edu

**Kevin Moran**
University of Central Florida
kpmoran@ucf.edu

**Ziyu Yao**
George Mason University
ziyuyao@gmu.edu

## Abstract

Despite remarkable advances in coding capabilities, language models (LMs) still struggle with simple syntactic tasks such as generating balanced parentheses. In this study, we investigate the underlying mechanisms behind the persistence of these errors across LMs of varying sizes (124M–7B) to both understand and mitigate the errors. Our study reveals that LMs rely on a number of components (attention heads and FF neurons) that independently make their own predictions. While some components reliably predict correct answers across a generalized range of inputs (i.e., implementing "sound mechanisms"), others are less reliable and introduce noise by promoting incorrect tokens (i.e., implementing "faulty mechanisms"). Errors occur when the faulty mechanisms overshadow the sound ones and dominantly affect the predictions. Motivated by this insight, we introduce RAS-TEER, a steering method to systematically identify and increase the contribution of reliable components for improving model performance. RASTEER substantially improves performance on balanced parentheses tasks, boosting accuracy of some models from $0\%$ to around $100\%$, without impairing the models' general coding ability. We further demonstrate its broader applicability in arithmetic reasoning tasks, achieving performance gains of up to around $20\%$.[1]

## 1 Introduction

Recent years have seen remarkable progress in the code generation capabilities of language models (LMs), driven by an increase in model size, training data, and improvements in overall training methodologies [17, 2, 27, 24, 15, 36, 2, 40]. Yet, despite these advances, LMs continue to struggle with basic syntactic tasks such as generating balanced parentheses and correct indentation [12, 45]. The failure of LMs to perform these seemingly simple tasks stands in stark contrast to their improved performance on more complex coding benchmarks [21, 9, 4], raising an important interpretability question: *How do LMs internally compute predictions for syntactic tasks and why do these computations sometimes fail?*

In this work, we seek to understand this failure by investigating the internal mechanisms of seven LMs, ranging in size from 124M to 7B parameters, while they perform the *balanced parentheses*

---

[1]The source code and dataset for the paper is available at https://github.com/Ziyu-Yao-NLP-Lab/failure-by-interference

39th Conference on Neural Information Processing Systems (NeurIPS 2025).

*task*, the task of predicting the correct number of closing parentheses in code statements. Recent work has attempted to reverse-engineer the full mechanisms of LMs [30, 33, 5]; however, even simple tasks were found to demand a bag of rather complicated computations [28, 25], which makes the bottom-up reverse engineering challenging. In our work, we instead understand the failure of an LM in a top-down manner, where we look for LM components, including attention heads and feed-forward (FF) neurons, that directly contribute to the final logit calculation of an LM. Our study shows that LMs employ a set of components, each with varying generalizability and reliability, to perform the balanced parentheses task. While most such components demonstrate high accuracy only within a narrow range of inputs and add noise in others, we still identify a rare set of highly effective and generalizable components. For instance, we find that a single attention head (L30H0) in CodeLlama-7b [36] outperforms the full model on our synthetic balanced parentheses dataset, highlighting the presence of strong, underleveraged mechanisms within the model. Key insights we derive from these findings are: (1) *LM doesn't rely on a single mechanism to make prediction but many mechanisms with varying levels of reliability*, and (2) *LMs do not fail due to the absence of sound mechanisms, but rather due to the presence of too many faulty mechanisms that introduce noise and overshadow the sound ones.*

Building on these interpretability insights, we propose RASTEER, an approach that RAnks LM components based on their reliability and STEERs generation by increasing the contribution of more reliable components to the final logits, for improving model performance. Despite the existing work in LM steering [35, 22], none has explored steering for tasks that are multi-class, position-sensitive, and have no clear "steering directions", as the balanced parentheses task. Applying RASTEER led to dramatic performance improvements across all seven models we studied, boosting accuracy on the balanced parentheses task from 0% to 100% for some models. To assess the broader impact of RASTEER, we evaluate whether steering for the balanced parentheses task affects the general code generation capabilities of models. On the HumanEval benchmark [9], we find that RASTEER preserves performance and even yields a modest improvement of 5.49% for Llama2-7b [17]. Finally, we also show the effectiveness of RASTEER beyond balanced parentheses tasks by applying it to an arithmetic reasoning task, where it achieves a performance gain of up to 20.25% for Pythia-6.9b [6].

## 2 Preliminaries

### 2.1 Background: Transformer-based Language Models

Transformer-based LM [43] maps an input sequence of tokens $X = (x_1, \ldots, x_n)$ to a probability distribution over the vocabulary $\mathcal{V}$ and predicts the next token $x_{n+1}$, typically by sampling or selecting the token with the highest probability. Initially, each input token $x$ is mapped to an embedding vector using a learned embedding matrix $W_E \in \mathbb{R}^{|\mathcal{V}| \times d}$, where $d$ is the model dimension, combined with positional encoding. The resulting representation $\mathbf{r}^0$ initializes the model's *residual stream*, which is refined sequentially across layers. At each layer $\ell \in \{1, \ldots, L\}$, the residual stream representation is updated sequentially by two sub-layers: a multi-head self-attention (MHSA) sub-layer followed by a feed-forward (FF) sub-layer:

$$\tilde{\mathbf{r}}^\ell = \mathbf{r}^\ell + \text{MHSA}^\ell(\text{LayerNorm}(\mathbf{r}^\ell)), \quad \mathbf{r}^{\ell+1} = \tilde{\mathbf{r}}^\ell + \text{FF}^\ell(\text{LayerNorm}(\tilde{\mathbf{r}}^\ell)). \tag{1}$$

**Multi-head Self-Attention (MHSA)** The MHSA sub-layer consists of multiple attention heads, indexed by head $h$ and operating in parallel. Each attention head performs a distinct computation that contributes additively to the residual stream. Specifically, the $h$-th attention head at layer $\ell$ computes:

$$\mathcal{H}^{\ell,h} = \text{Attn}(Q^{\ell,h}, K^{\ell,h})V^{\ell,h}W_O^{\ell,h}, \tag{2}$$

where $Q^{\ell,h} = \mathbf{r}^\ell W_{Query}^{\ell,h}$, $K^{\ell,h} = \mathbf{r}^\ell W_{Key}^{\ell,h}$, and $V^{\ell,h} = \mathbf{r}^\ell W_{Value}^{\ell,h}$ are the query, key, and value matrices computed from the input $\mathbf{r}^\ell$ using learned projection matrices $W_{Query}^{\ell,h}, W_{Key}^{\ell,h}, W_{Value}^{\ell,h} \in \mathbb{R}^{d \times d_{\text{head}}}$ and $W_O^{\ell,h} \in \mathbb{R}^{d_{\text{head}} \times d}$ are learned projection matrices specific to head $(\ell, h)$, and $d_{\text{head}}$ is the dimensionality per head.

**Feed-forward (FF) Sub-layer** The FF sub-layer at each layer $\ell$ consists of a two-layer feed-forward network:

$$\text{FF}^\ell(\tilde{\mathbf{r}}^\ell) = \sigma(\tilde{\mathbf{r}}^\ell W_K^\ell)W_V^\ell = \sum_{i=1}^{d_{\text{ff}}} \sigma(\tilde{\mathbf{r}}^\ell k_i^\ell)v_i^\ell = \sum_{i=1}^{d_{\text{ff}}} m_i^\ell v_i^\ell, \tag{3}$$

where $k_i^\ell \in \mathbb{R}^d$ is a column of the input projection matrix $W_K^\ell \in \mathbb{R}^{d \times d_{\mathrm{ff}}}$, and $v_i^\ell \in \mathbb{R}^d$ is a column of the output projection matrix $W_V^\ell \in \mathbb{R}^{d_{\mathrm{ff}} \times d}$. The bias term is omitted for simplicity. The activation function $\sigma(\cdot)$ is typically a non-linearity such as GeLU or ReLU. We follow Geva et al. [16] to decompose the computation, where $v_i^\ell$ is an *input-independent* parameter, which is referred to as an *FF neuron* in this work, and $m_i^\ell = \sigma(\tilde{\mathbf{r}}^\ell k_i^\ell)$ is an *input-dependent coefficient* representing the activation strength of the FF neuron.

After the final layer $L$, the model computes unnormalized logits over the vocabulary using the last-token residual stream output: $\mathrm{logits} = \mathbf{r}_n^L W_U$, where $W_U \in \mathbb{R}^{d \times |\mathcal{V}|}$ is the unembedding matrix.

## 2.2 LMs Failed in Naive Syntactic Code Completion

Recent empirical studies [12, 45] have found that a subset of LM code generation errors stem from failures to accurately complete basic syntactic structures like the balanced parentheses task, an issue that persists even in state-of-the-art models such as GPT-4 [2] and Phi-3 [1]. To systematically study this LM behavior, we decompose the task of balanced parentheses as a collection of sub-tasks, determined by how an LM tokenizer processes sequences of $N$ closing parentheses. [2] Specifically, every LM tokenizer in our study represents one, two, three, and four closing parentheses as single tokens, while sequences with $N > 4$ are split into multiple tokens of these one to four closing parentheses tokens. As a result, we define four sub-tasks within the broader balanced parentheses task, corresponding to $N = \{1, 2, 3, 4\}$, and synthesize a separate dataset for each. Specifically, we synthesize dataset using the following template for each sub-task:

- **One-Paren:** #print the string $\{num\}$\nprint($\{num\} \to$ )
- **Two-Paren:** #print the string $\{num\}$\nprint(str($\{num\} \to$ ))
- **Three-Paren:** #print the string $\{num\}$\nprint(str(str($\{num\} \to$ )))
- **Four-Paren:** #print the string $\{num\}$\nprint(str(str(str($\{num\} \to$ ))))

For each sub-task, we generate 350 training, 150 dev, and 150 test examples, each consisting of the input prompts created by randomly selecting a numeric value $\{num\}$ from the range 100 to 999 and ground-truth output tokens. We select seven models—GPT-2 Small [31], GPT-2 Medium [31], GPT-2 Large [31], GPT-2 XL [31], CodeLlama-7b [36], Llama-2 7b [40], and Pythia-6.9b [6]—based on a combination of factors: model size (ranging from 117M to 7B parameters), model family (including both general-purpose base models and Code LMs), and performance diversity (capturing a broad spectrum of accuracy from 0% to 100% on our synthetic dataset). Specifically, all models had 100% accuracy for the one and two-paren task. However, most models exhibit low accuracy on the three-paren and four-paren sub-tasks, particularly all GPT-2 models had 0% accuracy on the four-paren sub-task. Full results can be found in Appendix A.

## 3 Understanding LMs in Making Balanced Parentheses Errors

### 3.1 Overview

To understand why an LM makes (in)correct predictions, we investigate LM components (i.e., attention heads and FF neurons) that *directly contribute to the final logit of the model* from the last-token position. By focusing on these components, we abstract away the need to analyze the full underlying mechanisms, which can be highly labor-intensive and complex. Crucially, since all internal computations must ultimately influence the model's output through these final components, examining them can still provide necessary insights to understand when and why a prediction goes wrong. Specifically, we will apply Algorithm 1 to identify components that selectively promote the correct token over the distractors, and Algorithm 2 to identify components that promote the correct token with a thresholded strength. We consider LM components that selectively promote correct tokens with reasonable strength across generalized contexts as *reliable contributors* that implement *sound mechanisms*, and others as *unreliable contributors* implementing *faulty mechanisms*.

The two algorithms both utilize the logit lens technique [29], which projects the corresponding *component activation* to the vocabulary space. For attention heads, the component activation $\mathbf{h}_c$ is

---

[2]Empirically, when an LM does not generate parentheses following the way how its tokenizer works (e.g., predicting ")) " first and expecting another ")) ", rather than predicting ")))) "), it can hardly succeed.

**Algorithm 1** Measuring Task Correctness for LM Components

**Require:** Component activation $\mathbf{h}_c \in \mathbb{R}^d$, unembedding matrix $W_U \in \mathbb{R}^{d \times |\mathcal{V}|}$, ground-truth token $t \in \mathcal{V}$, and distractor tokens $\mathcal{T}_{neg} \subset \mathcal{V}$
1: Compute logits: $\mathbf{l}_c \leftarrow \mathbf{h}_c^\top W_U \in \mathbb{R}^{|\mathcal{V}|}$
2: **if** $\mathbf{l}_c[t] \geq \max(\mathbf{l}_c[\mathcal{T}_{neg}])$ **then**
3:     **return** True       ▷ Marked as correct
4: **else**
5:     **return** False
6: **end if**

**Algorithm 2** Labeling Token Promotion for LM Components

**Require:** Component activation $\mathbf{h}_c \in \mathbb{R}^d$, unembedding matrix $W_U \in \mathbb{R}^{d \times |\mathcal{V}|}$, target token $t \in \mathcal{V}$, and promotion threshold $\tau \in [0, 1]$
1: Compute logits: $\mathbf{l}_c \leftarrow \mathbf{h}_c^\top W_U \in \mathbb{R}^{|\mathcal{V}|}$
2: **if** $\mathbf{l}_c[t] \geq \tau \cdot \max(\mathbf{l}_c)$ **then**
3:     **return** True       ▷ token $t$ is labeled
4: **else**
5:     **return** False
6: **end if**

$\mathcal{H}^{\ell,h}$; for FF neurons, $m_i^\ell v_i^\ell$. Given the large number of FF neurons in each model (e.g., 131,072 in CodeLlama-7b), we perform a static pre-filtering step to exclude neurons that are unlikely to affect the target tokens. Specifically, we follow prior work [16] to interpret an FF neuron by projecting its parameter to the vocabulary space using the unembedding matrix, i.e., $v_i^\ell W_U$. We then retain only neurons whose projections include at least one of the four closing-parenthesis tokens among their top-50 or bottom-50 logit-ranked tokens.

### 3.2 LMs Developed Mechanisms of Varying Levels of Generalizability

We start with measuring the *task accuracy* of each LM component on each sub-task. Specifically, for every input prompt on a sub-task training set, we follow Algorithm 1 to measure if the component's logit projection yields the highest value to the ground-truth token *among all four token choices*. In other words, we check if the component can correctly promote the correct token more than the incorrect ones. Based on the correctness counts, we calculate the accuracy of the component for each sub-task. We visualize the accuracy distributions of the attention heads of CodeLlama in Appendix B.1. We observe that most models consistently have more high-accuracy (e.g., greater than 0.7) components for the one-paren sub-task, with the count gradually decreasing for the two-paren, three-paren, and four-paren sub-tasks.[3] This trend closely mirrors the overall performance of the models on each respective sub-task.

Based on the observed accuracy distributions, we decide to group LM components based on the number of sub-tasks in which they achieve high accuracy, using 0.7 as a threshold. Specifically, if a component achieves at least 0.7 accuracy in more than one sub-task, we interpret it as implementing (i.e., serving as the prediction head of) a "sound mechanism" that generalizes across sub-tasks. Table 1 reports the number of attention heads and FF neurons that generalize to different numbers of sub-tasks. Our analysis reveals that most components are specialized, attaining high accuracy on only a single sub-task, while a smaller subset generalizes effectively across multiple sub-tasks. These results indicate that *LMs implement a diverse set of mechanisms to solve each sub-task, with varying degrees of generalizability*.

Table 1: Number of LM components (attention heads and FF neurons) that have high accuracy ($\geq 70\%$) across different numbers of sub-tasks.

| Model (*# heads, # neurons*) | Attention Heads | | | | FF Neurons | | | |
|---|---|---|---|---|---|---|---|---|
| | 1 task | 2 tasks | 3 tasks | 4 tasks | 1 task | 2 tasks | 3 tasks | 4 tasks |
| GPT-2 Small *(144, 9,216)* | 11 | 1 | 0 | 0 | 155 | 0 | 0 | 0 |
| GPT-2 Medium *(384, 24,576)* | 10 | 2 | 0 | 0 | 235 | 0 | 0 | 0 |
| GPT-2 Large *(720, 46,080)* | 18 | 4 | 0 | 0 | 365 | 1 | 0 | 0 |
| GPT-2 XL *(1,200, 76,800)* | 38 | 3 | 1 | 0 | 626 | 3 | 0 | 0 |
| CodeLlama-7b *(1,024, 131,072)* | 27 | 6 | 0 | 1 | 852 | 6 | 0 | 0 |
| Llama2-7b *(1,024, 131,072)* | 18 | 2 | 0 | 0 | 391 | 5 | 0 | 0 |
| Pythia-6.9b *(1,024, 131,072)* | 23 | 4 | 0 | 0 | 1893 | 2 | 0 | 0 |

---

[3]The accuracy threshold of 0.7 was selected based on empirical observations of accuracy distributions across sub-tasks (Figure 4). We found that this value effectively separates components with high accuracy in at least one sub-task from others.

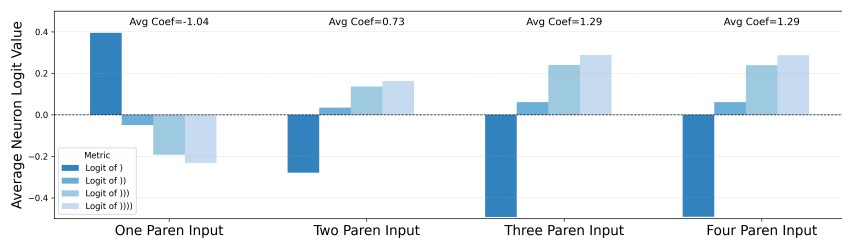

Figure 1: Average logit values and coefficients of FF neuron L19N11 of CodeLlama when the input prompts demand one-, two-, three-, and four-paren closing tokens.

**Generalization Capability of Attention Heads** We further look into attention heads that generalize across multiple sub-tasks. These heads are listed in Table 4. We observe that attention heads with stronger generalization tend to emerge in deeper layers of the model. Notably, for CodeLlama-7b, attention head 0 at layer 30 (L30H0) achieved almost $100\%$ accuracy across all sub-tasks, which was even better than the full model (96.00% on average). In Table 5, we see that the generalizable attention head primarily attends to the first function name token (L30H0) or open parenthesis (L30H16) that has yet to be closed across all of the sub-tasks. In comparison, the attention scores of a non-generalized attention head are largely spread across different prompt tokens, with the major attention being placed on the begin-of-sentence token (omitted in the visualization).

**Generalization Capability of FF Neurons** Prior work [11, 32] interpreted an FF neuron mainly by looking at only the top-k tokens sorted by their logit scores under the input-independent projection of $v_i^\ell W_U$. However, contrary to this conjecture, our analysis reveals that *LMs do not rely solely on the top-k logits—rather, they also utilize bottom-k logits through a dual-sign mechanism.* Specifically, neurons apply *positive* coefficients to promote top-k tokens and *negative* coefficients to suppress them or, equivalently, to promote bottom-k tokens. This mechanism allows a single neuron to support two distinct sub-tasks when its input-independent projection contains relevant tokens at both extremes. For example, neuron 11 in layer 19 (L19N11) of CodeLlama has ")" among its bottom-10 tokens, and ")))", and "))))" among its top-50 tokens, after the input-independent projection. As shown in Figure 1, the neuron promotes the bottom token ")" for one-paren inputs by assigning it a negative coefficient (on average, $-1.04$), but promotes the top token "))))" for four-paren inputs by assigning it a positive coefficient (on average, $1.29$). Being able to "flip" the coefficient depending on the input prompts allows this neuron to promote the correct tokens generalizably across both the one-paren and the four-paren sub-tasks. However, because both ")))" and "))))" are ranked as top tokens for L19N11, when the neuron promotes "))))", it inevitably also promotes ")))". We term this phenomenon as *noisy promotion*, meaning that the FF neuron has to promote the ground-truth and the distractor tokens at the same time. This observation highlights both an *architectural constraint* and an *adaptation* developed with FF neurons—*because FF neurons can only rank tokens on either the top side or the bottom side of its parametric memory, it can generalize to at most two sub-tasks; however, it attempts to overcome this limitation by developing a noisy promotion strategy.*

### 3.3 LMs Predict via Noisy Promotion and Low Selectivity

To further understand the "noisy promotion" effect of LM components, we conduct the second analysis to look at the *recall* and *precision* of an LM component's promotion. Unlike the previous analysis, which focuses on whether an LM component comparatively ranks the correct token with a higher logit than the other three distractors ($\mathcal{T}_{neg}$), this analysis checks the absolute logit value projected by an LM component to each answer token. Specifically, recall measures whether a component promotes the correct answer token for the associated inputs (irrespective of whether the distractors are promoted), and precision measures whether the component promotes a token only when the token is the true answer.

We use Algorithm 2 to identify which tokens each LM component promotes, based on a fixed *promotion threshold* $\tau$, set to $0.5$ in our analysis (see Appendix B.2 for details on setting the promotion threshold to $0.5$). To compute recall and precision for each sub-task, we construct a balanced dataset containing equal numbers of positive and negative examples, using the same training set introduced in Section 2. For instance, in the one-paren sub-task, half of the examples require ")"

as the correct token sampled from the one-paren train set, while the remaining examples are sampled evenly from the other three sub-tasks.

Figure 2 presents the precision–recall scatter plots, averaged across all sub-tasks, for all attention heads and FF neurons in CodeLlama-7b and GPT-2 Small; results for other models are shown in Appendix B.3. Our analysis reveals that most components exhibit both low precision and low recall. A small subset achieves high recall, but notably, we observe an absence of components with high precision—highlighting a widespread lack of *selectivity* across models. This observation suggests that LMs rely on a large number of components and make heavy use of noisy promotion, where components often activate for both correct and incorrect tokens—boosting the correct token more strongly, but not exclusively. Consequently, predictions emerge from the aggregate effect of many low-selectivity components, rather than from a small number of highly precise ones.

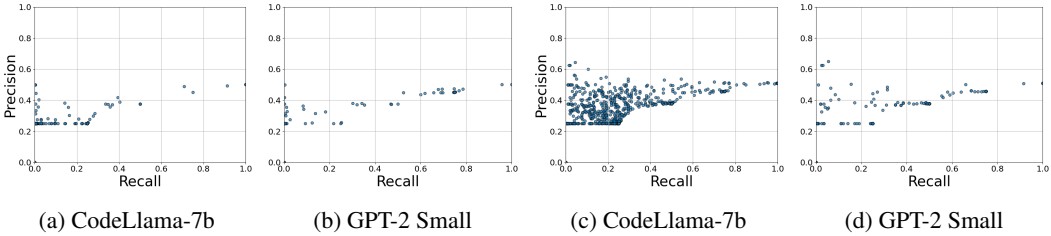

|        (a) CodeLlama-7b        |        (b) GPT-2 Small        |        (c) CodeLlama-7b        |        (d) GPT-2 Small        |

Figure 2: Scatter plots of average precision vs. recall across all sub-tasks for attention heads (left two) and FF neurons (right two) of CodeLlama-7b and GPT-2 Small.

## 4 Ranking and Steering LM Components for Performance Enhancement

Our analysis in Section 3 indicates that LMs developed both sound and faulty mechanisms for the balanced parentheses tasks, and that they make predictions via a noisy promotion strategy. We hypothesize that the errors of balanced parentheses do not stem from the absence of sound mechanisms, but rather from their influence being *overshadowed* by the faulty ones that introduce noise into the model's computation. Inspired by the hypothesis, we propose RASTEER, an approach aiming to improve an LM's performance on the balanced parentheses task by first RAnking LM components based on the soundness of their mechanisms and then STEERing these components to augment their effect, so as to get rid of the mechanism overshadowing.

### 4.1 RASTEER: Ranking and Steering LM Components

We rank the LM components in the following order. First of all, we group LM components based on their generalizability (Section 3.2) and sort them based on the number of sub-tasks they generalize to in a descending order. Then, within each generalizability group, we further sort components based on their promotion effect averaged over all the sub-tasks (Section 3.3). In experiments, we consider recall, precision, and F1 as the metrics and decide the most effective one based on the experimental results. Like in our analysis, the sorting process utilized a training set for each sub-task.

Given a sorted list of LM components, we perform LM steering to increase the impact of the top-$k$ components on the final prediction. Specifically, for each selected component $c$, we scale its activation $\mathbf{h}_c$ by a multiplier $\alpha \in [1.1, 2.0]$ before adding it to the residual stream.

### 4.2 A Strong Baseline: Ranking LM Components from Circuit Discovery

In contrast to our top-down approach of only finding LM components that have strong contributions to a model's prediction, recent research on Mechanistic Interpretability (MI) [33, 5, 14] takes an bottom-up approach to identify a complete mechanism (particularly, circuits [44, 18, 28]) for LM behaviors Given its promise, we add a circuit baseline. Specifically, we discover one circuit for each sub-task following the activation patching of Nikankin et al. [28], which focuses on localizing causally important attention heads while retaining all FF layers.[4] We then rank each attention head based on its patching effect averaged over sub-tasks. Steering is only performed to attention heads. We leave details of our circuit discovery in Appendix C.

---

[4]We do not localize both attention heads and FF neurons due to the prohibitively high computational cost.

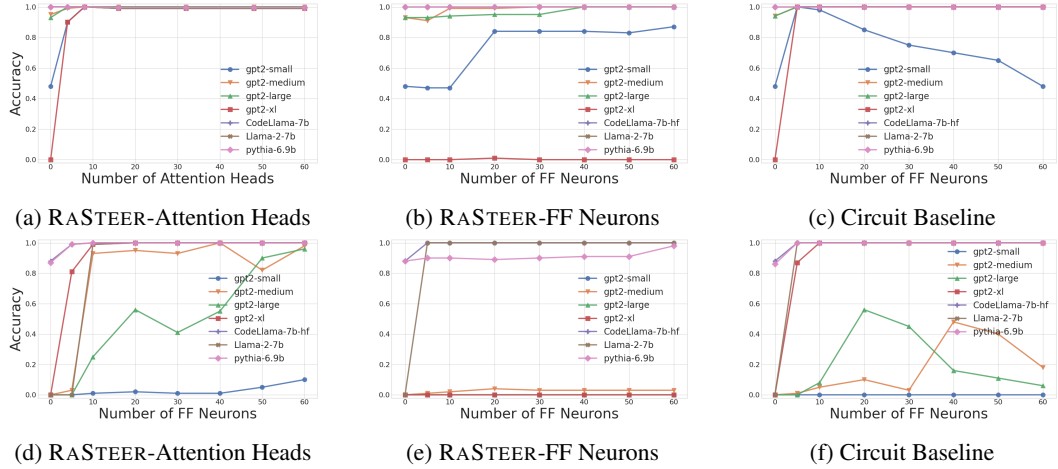

(a) RASTEER-Attention Heads     (b) RASTEER-FF Neurons     (c) Circuit Baseline

(d) RASTEER-Attention Heads     (e) RASTEER-FF Neurons     (f) Circuit Baseline

Figure 3: Performance of RASTEER (steering either attention heads or FF neurons) and the circuit baseline on the three-paren (top) and the four-paren (bottom) sub-tasks. When zero heads or neurons are steered, it shows each model's raw performance without steering. Results of RASTEER on one-paren and two-paren sub-tasks and when steering both attention heads and FF neurons are provided in Appendix D.1.

## 5 Experiments

### 5.1 Experimental Setup

**Dataset** We mainly evaluate our approach on the *test set* of balanced parentheses for each sub-task, which consists of 150 examples drawn from the same synthetic configuration but disjoint from the training set, as described in Section 2. In addition, to verify that the steering intervention does not degrade the model's general coding performance, we also evaluate our approach on *HumanEval* [9], a standard benchmark dataset for evaluating the coding capability of an LM. For both datasets, we report the accuracy of the model before and after steering.

**Approaches and Configuration** We experiment with three variants when applying RASTEER to steer (1) only attention heads, (2) only FF neurons, and (3) both attention heads and FF neurons. Notably, we promote the same set of components for all sub-tasks as we rank LM components based on an averaged effect over all sub-tasks; we expect steering to enhance all sub-tasks. The demand is also because, in practice, we would not foresee the category of sub-task that an upcoming input falls into. Similarly, we steer attention heads across all token positions, since we do not assume prior knowledge of the position at which the model must generate a closing parenthesis. In each variant, we vary the number of top-ranked components to steer and observe its impact on the model performance. For the circuit baseline, since we only calculate effect scores for attention heads, the steering experiments were conducted on attention heads only. For all steering experiments, we use the dev set to determine the optimal scaling multiplier ($\alpha$) for each model. Model inference is performed via greedy decoding for stability.

### 5.2 Main Experimental Results

**RASTEER provides dramatic improvement on three-paren and four-paren sub-tasks** In Figure 3, we present the effect of RASTEER on the three-paren and four-paren sub-tasks, when steering the top attention heads or the top FF neurons. On the one-paren and two-paren sub-tasks, all models achieved almost perfect accuracy pre-steering and were not impacted by steering. Our approach, by promoting more reliable components at the top, leads to dramatic performance gains on the balanced parentheses tasks. For instance, GPT-2 XL initially achieved $0\%$ accuracy on three and four-paren sub-tasks, yet promoting only the top-10 attention heads results in an improvement to $\sim 100\%$ accuracy. Similar trends were observed across all models: promoting the top-60 attention heads leads to $\sim 100\%$ accuracy of models on all sub-tasks, with the exception of GPT-2 Small on the four-paren sub-task (see Appendix D.2 for analysis of why RASTEER fails to improve GPT-2 Small in the four-paren

sub-task). Interestingly, we observe that larger models generally require fewer component promotions to achieve strong performance. For example, in the four-paren sub-task, only GPT-2 Small and Medium required more than the top-10 attention heads for substantial improvements, while all other models achieve $\sim 100\%$ accuracy by promoting just the top-10 attention heads.[5]

**Attention heads promotion yielded better performance than promoting FF neurons** We observed that promoting only the attention heads achieved performance comparable to joint promotion of both the attention heads and the FF neurons, while promoting only FF neurons resulted in little to no improvement, particularly for smaller models. For example, in GPT-2 XL, promoting sixty FF neurons had no effect on the three-paren sub-task, whereas promoting just five attention heads improved model performance from $0\%$ to $100\%$. A similar trend was observed across all GPT-2 models on the four-paren sub-task. We posit that attention heads are more effective for steering because they are not structurally constrained in the same way as FF neurons, which appear to generalize to only a limited number of sub-tasks, as discussed in Section 3.2. Additionally, FF neurons may require coordination with other components to influence predictions, rather than contributing meaningfully in isolation. Based on these findings, we restrict our subsequent experiments to the attention head steering.

**RASTEER with components ranked by F1-score has the best performance** We ranked components using three reliability metrics: recall, precision, and F1-score. We show the results based on F1-score in Figure 3 and others in Appendix D.3. Among these, F1-score consistently yielded the best overall performance. Between recall and precision, recall outperforms precision—aligning with our earlier analysis in Section 3, which suggests that LMs tend to rely more on noisy promotion rather than highly selective, precise components.

**RASTEER outperforms the circuit baseline** As shown in Figure 3, RASTEER consistently outperforms the circuit baseline across models and tasks. While the circuit baseline yields comparable improvements in models with over a billion parameters, it proves ineffective for smaller models. For example, in the four-paren sub-task, all GPT-2 models—except GPT-2 XL—failed to benefit from promoting the top-60 attention heads ranked by circuit analysis. Furthermore, performance even declined as more heads were promoted, suggesting that component rankings derived from circuit analysis do not reliably translate into effective steering strategies for improving model performance. To understand the discrepancy, we further analyze the reasons why the circuit baseline is effective for larger models (over 1B parameters) but unstable for smaller ones. For larger models, we first examine the differences in attention heads selected by the circuit baseline and RASTEER. Our analysis shows that the degree of overalp vary between 20–100% as shown in Figure 9, and steering these overlapping heads yields mixed results—effective for Llama-2 7b and CodeLlama-7b, but not in Pythia-6.9b, indicating redundancy in useful mechanisms. For smaller models, we find that the instability mainly arises from steering attention heads that do not directly contribute to the final logits but are instead involved in intermediate computations, such as information transfer and feature extraction. Steering these additional heads drastically reduces performance, demonstrating that not all functionally relevant components are beneficial for steering. We include further discussion in Appendix D.4.

### 5.3 HumanEval Results

To ensure that our LM steering for the balanced parentheses task does not adversely impact broader code generation capabilities, we evaluate the post-steering performance on the HumanEval benchmark [9] using CodeLlama-7b, Llama2-7b, and Pythia-6.9b. We did not experiment with GPT-2 models because they had 0% accuracy on the HumanEval benchmark. Steering the top-20 attention heads—selected from the parentheses task and scaled with a multiplier range of $[1.1, 2.0]$, did not degrade HumanEval performance and even led to a 5.49% improvement for Llama2-7b: CodeLlama ($30.48\% \rightarrow 29.87\%$), Llama2 ($11.58\% \rightarrow 17.07\%$), Pythia ($10.36\% \rightarrow 10.97\%$). However, extending steering beyond the top-20 heads resulted in performance degradation: CodeLlama-7b and Pythia-6.9b experienced a modest decline of 1–1.5% when up to 60 heads were promoted, whereas Llama2-7b maintained a slight gain with best performance while promoting just top-20 attention heads. These results suggest that *targeted promotion of a small set of reliable components*

---

[5]We have also experimented with the more recent Qwen2.5-3b model [38] and shown that RaSTEER (applied to top-20 attention heads) can improve its accuracy on Three- and Four-Paren subtasks from 92.66% to 98.00% and from 46.66% to 77.33%, respectively.

*can improve task-specific performance without compromising general capabilities, while broader interventions may lead to diminishing returns or adverse effects.*

### 5.3.1 Does RaSteer Generalize to Arithmetic Reasoning?

We further assess RaSteer on an arithmetic task using three models: GPT-2 XL, Pythia-6.9b, Pythia-12b, and Llama3-8b [13].[6] We consider a two-operand arithmetic reasoning task, for which we construct a dataset comprising 750 training, 350 dev, and 350 test set examples. Each input prompt consists of four tokens: the first operand, an operator $(+, -, \times, \div)$, the second operand, and an equals sign. To construct the dataset, we randomly sample integers in the range $[0, 500]$ for both operands and ensure that the resulting answer also falls within this range to ensure single-integer tokenization. Because the task does not have sub-task categorization similar to the balanced parenthese task, when applying RaSteer, we only ranked LM components based on their promotion effect on the training set, following Algorithm 2. We chose recall as the promotion effect metric as we observed a similar noisy promotion strategy in this task.

As shown in Table 2, RaSteer yields performance improvements across most arithmetic operations for all three models, with the largest gain of 20.25% observed in Pythia-6.9b for multiplication. These results show that our method can potentially generalize beyond the balanced parentheses task and also indirectly support our central insights from Section 3: *LMs rely on a large number of components with varying reliability, generating predictions through the noisy token promotion of these components.*

Table 2: Performance of RaSteer when steering top-30 attention heads sorted by recall on two-operand arithmetic tasks

| Model | Addition | Subtraction | Multiplication | Division |
|---|---|---|---|---|
| GPT-2 XL | $0.00\% \rightarrow 0.00\%$ | $0.00\% \rightarrow 0.00\%$ | $17.50\% \rightarrow 17.50\%$ | $19.66\% \rightarrow 23.66\%$ |
| Pythia-6.9b | $25.41\% \rightarrow 31.66\%$ | $0.00\% \rightarrow 2.50\%$ | $14.41\% \rightarrow 34.66\%$ | $19.91\% \rightarrow 34.75\%$ |
| Pythia-12b | $17.33\% \rightarrow 24.66\%$ | $6.00\% \rightarrow 7.33\%$ | $8.00\% \rightarrow 21.00\%$ | $27.00\% \rightarrow 36.33\%$ |
| Llama3-8b | $82.33\% \rightarrow 84.33\%$ | $87.91\% \rightarrow 88.00\%$ | $74.08\% \rightarrow 80.08\%$ | $78.16\% \rightarrow 79.91\%$ |

## 6 Related Work

**LMs for Code Generation and Syntactic Failures** Recent advances in LMs have led to significant improvements in their code generation capability [17, 2, 27, 24, 15, 36]. However, LMs are still prone to a range of semantic and syntactic errors, including balanced parentheses, and indentations [12, 45]. To investigate these findings, our work focuses on the balanced parentheses task as a representative task to study the internal mechanisms of LMs for the syntactic task. Furthermore, we also explore whether we can leverage understanding from our interpretability study to mitigate these failures without harming overall model performance on broader code generation tasks.

**Mechanistic Interpretability (MI)** MI is a subfield of interpretability that aims to reverse-engineer the algorithms learned by a model [25, 7, 14, 30, 32]. In our study, we employ techniques and insights from MI to identify and study LM components that implement the balanced parentheses task. However, in our work, we do not aim to fully reconstruct the underlying algorithm or circuit responsible for this behavior. This is primarily because our analysis in Section 3 indicated that the model does not rely on a single mechanism, but rather on a large number of components, making it impractical to interpret each mechanism in detail, in line with recent work findings that LMs often employ multiple mechanisms for tasks such as arithmetic reasoning [28] and factual recall [10].

**LM Generation Steering** Steering LM generation has recently emerged as a popular lightweight approach for controlling model behavior at inference time [35, 41, 47, 26]. Most approaches use *steering vectors* to elicit high-level traits like honesty [47], truthfulness [22], sycophancy [35], or sentiment [39], and have been applied to enhance capabilities [46, 26, 42] and support red-teaming [3, 34]. In our work, we do not use the steering vectors approach, as it is not clear what vectors to look for a multi-class, position-sensitive task like balancing parentheses. Instead, we

---

[6]Other GPT-2 models are excluded due to their inability to perform the task, while CodeLlama-7b and Llama2-7b are omitted because they tokenize a complete integer into multiple tokens of digits, making the token prediction-based analysis difficult. Following Nikankin et al. [28], we avoid such models.

focus on identifying and promoting specific model components that contribute reliably to correct predictions. While activation steering has been extensively studied, steering generation through the promotion of individual model components remains relatively underexplored [16, 23].

## 7 Limitations and Conclusion

In this work, we study the mechanisms for why LMs make errors in simplistic balancing parentheses tasks and also propose a steering approach to improve them. While RASTEER substantially improves LM performance on both balanced parentheses and arithmetic reasoning tasks, our study conducted using synthetic data may have exaggerated its effectiveness. Additionally, we analyze LM failures on the balanced-parentheses task, which can be cleanly categorized into four distinct sub-tasks depending on how the LM tokenizes different numbers of closing parentheses. However, this decomposition may not generalize to all tasks, especially those with a much larger set of possible output tokens. Although we demonstrate that RASTEER remains effective when the output space expands, as in arithmetic reasoning, there may exist other tasks—such as general reasoning—where the output space is even larger or less well-defined, potentially limiting the approach's applicability. Besides the task setup, our approach assumes that LMs follow a simple additive motif, where the final logit is formed by simply adding the contributions from individual components. While this assumption is also supported by several prior findings [10, 14] and further reinforced by our results with RASTEER, it may overlook non-additive mechanisms—such as those that actively suppress noise. Investigating such dynamics remains an important direction for future work. Our method also relies on simple heuristics: components are ranked using recall, precision, and F1-score, and promoted via fixed scalar multipliers. Future work could explore more sophisticated techniques for both ranking and promotion. Lastly, rather than reconstructing full mechanisms or circuits, we focus on steering components that directly influence the final logit. We believe that this form of functional interpretability, centered on understanding high-impact LM components for practical applications, remains underexplored.

## Acknowledgments and Disclosure of Funding

This project was sponsored by the National Science Foundation (Award Number 2311468/2423813). The project was also supported by GPU resources provided by the Office of Research Computing at George Mason University (URL: `https://orc.gmu.edu`) and funded in part by grants from the National Science Foundation (Award Number 2018631).

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

# A  Model Performance on Balanced Parentheses Task

Table 3: Accuracy of LMs on the balanced parenthesis task, LM when the ground-truth token includes one, two, three, and four closing parentheses, respectively.

| Model | One-Paren | Two-Paren | Three-Paren | Four-Paren |
|---|---|---|---|---|
| GPT-2 Small | 100.00% | 100.00% | 49.00% | 0.00% |
| GPT-2 Medium | 100.00% | 100.00% | 93.00% | 0.00% |
| GPT-2 Large | 100.00% | 100.00% | 88.00% | 0.00% |
| GPT-2 XL | 100.00% | 100.00% | 0.00% | 0.00% |
| Llama2-7b | 100.00% | 100.00% | 100.00% | 1.00% |
| CodeLlama-7b | 99.00% | 100.00% | 98.00% | 87.00% |
| Pythia-6.9b | 100.00% | 100.00% | 100.00% | 83.00% |

# B  Additional Results of LM Components Analysis

## B.1  Accuracy Distributions of Attention Heads across Sub-tasks

The accuracy distribution of the attention heads for GPT-2 Small, GPT-2 Medium, GPT-2 Large, Llama2-7b, and Pythia-6.9b is shown in Figure 4.

## B.2  Selection of Promotion Threshold

To determine an appropriate promotion threshold, we initially selected 0.5 as a reasonable starting point for the analysis in Section 3, and later validated this choice through tuning on the development set for RASTEER. Specifically, we evaluated thresholds in the range $[0.4, 0.5, 0.6, 0.7, 0.8, 0.9]$ and found that 0.5 consistently achieved strong performance. For example, on the four-paren sub-task with GPT-2 Small (top-60 heads), 0.5 yielded 16.8% accuracy—outperforming other thresholds, which ranged between 5.3% and 14.0%. Notably, we found that the promotion threshold had less impact on larger models ($\geq$ 1B parameters), where promoting just a small number of reliable components is often sufficient for performance gains. In summary, we posit that the promotion threshold balances between false positives and false negatives: thresholds below 0.5 risk including too many components that do not meaningfully promote the target token, while thresholds above 0.5 risk excluding components that are genuinely helpful.

## B.3  Precision vs. Recall Analysis Across Models

Figure 5 shows the precision vs recall analysis across all sub-tasks for both attention head and neurons. For all models, most components exhibit both low precision and low recall. A small subset achieves high recall, but notably, we observe an absence of components with high precision—highlighting a widespread lack of *selectivity* across models.

## B.4  Analysis of Generalizable Attention Heads

In Table 4, we list the attention heads that generalize to one or more sub-tasks and FF neurons that generalize to more than one sub-task for each model. In Table 5, we present the attention visualization of three heads, two generalizable and one not.

Table 4: List of attention heads that generalize to one or more sub-tasks and FF neurons that generalize to more than one sub-task.

| Model | Generalizable Attention Heads | Generalizable FF Neurons |
|---|---|---|
| GPT-2 Small | L1H1 (1-paren), L1H4 (2-paren), L2H6 (1-paren), L4H5 (2-paren), L5H3 (2-paren), L6H3 (3-paren), L7H6 (4-paren), L8H5 (1-paren), L10H3 (2-paren), L10H5 (3-paren), L11H8 (1-paren), L9H10 (1-paren, 2-paren) | N/A |
| GPT-2 Medium | L1H7 (1-paren), L9H8 (3-paren), L13H10 (2-paren), L14H7 (2-paren), L16H0 (2-paren), L16H3 (4-paren), L19H3 (2-paren), L19H9 (3-paren), L21H1 (1-paren), L23H8 (3-paren), L17H5 (1-paren, 3-paren), L18H6 (1-paren, 2-paren) | N/A |
| GPT-2 Large | L1H3 (1-paren), L2H5 (1-paren), L3H11 (1-paren), L4H17 (1-paren), L5H17 (1-paren), L6H19 (1-paren), L8H11 (2-paren), L10H4 (3-paren), L11H3 (4-paren), L12H8 (4-paren), L12H15 (4-paren), L13H10 (4-paren), L14H3 (4-paren), L15H14 (2-paren), L18H11 (2-paren), L23H6 (2-paren), L27H3 (2-paren), L30H9 (4-paren), L22H15 (1-paren, 2-paren), L24H3 (1-paren, 4-paren), L25H4 (1-paren, 2-paren), L26H8 (2-paren, 3-paren) | L29N1354 (1-paren, 2-paren) |
| GPT-2 XL | L1H24 (1-paren), L2H13 (1-paren), L3H10 (1-paren), L4H2 (1-paren), L7H16 (4-paren), L7H22 (4-paren), L14H3 (4-paren), L14H18 (1-paren), L15H9 (3-paren), L16H22 (4-paren), L18H13 (4-paren), L18H23 (4-paren), L19H4 (2-paren), L19H6 (4-paren), L21H0 (4-paren), L21H5 (3-paren), L21H11 (4-paren), L22H13 (4-paren), L22H18 (4-paren), L23H3 (2-paren), L23H6 (4-paren), L24H15 (4-paren), L24H24 (2-paren), L25H0 (2-paren), L26H16 (3-paren), L26H18 (4-paren), L27H7 (2-paren), L27H10 (2-paren), L29H3 (4-paren), L29H12 (2-paren), L30H14 (2-paren), L31H18 (4-paren), L32H0 (1-paren), L33H11 (2-paren), L34H14 (2-paren), L35H22 (2-paren), L41H19 (2-paren), L42H5 (1-paren), L13H17 (1-paren, 4-paren), L32H1 (2-paren, 3-paren), L42H16 (2-paren, 3-paren), L30H16 (1-paren, 2-paren, 3-paren) | L36N1149 (1-paren, 3-paren), L36N5870 (1-paren, 3-paren) |
| CodeLlama-7b | L0H0 (1-paren), L1H6 (1-paren), L1H7 (4-paren), L8H29 (1-paren), L15H16 (1-paren), L16H12 (4-paren), L18H20 (1-paren), L18H21 (1-paren), L21H22 (2-paren), L22H4 (1-paren), L24H18 (1-paren), L24H23 (1-paren), L25H5 (1-paren), L26H6 (1-paren), L27H4 (3-paren), L28H22 (4-paren), L28H29 (3-paren), L29H7 (1-paren), L29H12 (1-paren), L29H29 (1-paren), L30H11 (1-paren), L30H13 (1-paren), L31H4 (1-paren), L31H8 (4-paren), L31H10 (1-paren), L31H14 (1-paren), L31H18 (1-paren), L17H17 (1-paren, 2-paren), L27H15 (1-paren, 2-paren), L27H24 (1-paren, 2-paren), L28H13 (1-paren, 2-paren), L29H0 (1-paren, 2-paren), L31H22 (3-paren, 4-paren), L30H0 (1-paren, 2-paren, 3-paren, 4-paren) | L19N11 (1-paren, 4-paren), L20N3998 (1-paren, 4-paren), L22N8326 (1-paren, 2-paren), L27N9695 (2-paren, 3-paren), L29N8515 (1-paren, 2-paren) |
| Llama2-7b | L1H30 (1-paren), L3H3 (1-paren), L17H17 (1-paren), L18H21 (1-paren), L20H25 (3-paren), L22H4 (1-paren), L24H18 (1-paren), L25H5 (2-paren), L25H24 (4-paren), L27H24 (2-paren), L28H13 (1-paren), L28H29 (4-paren), L29H0 (1-paren), L31H4 (1-paren), L31H5 (4-paren), L31H8 (2-paren), L31H21 (1-paren), L31H22 (4-paren), L27H15 (1-paren, 2-paren), L30H0 (2-paren, 3-paren) | L15N6063 (1-paren, 3-paren), L19N11 (1-paren, 4-paren), L20N3998 (1-paren, 4-paren), L27N5474 (2-paren, 4-paren), L30N9014 (1-paren, 2-paren) |
| Pythia-6.9b | L4H21 (1-paren), L9H20 (3-paren), L10H14 (1-paren), L10H21 (1-paren), L12H12 (3-paren), L13H7 (1-paren), L13H26 (2-paren), L14H9 (1-paren), L14H11 (2-paren), L15H9 (1-paren), L16H12 (1-paren), L17H10 (1-paren), L18H6 (4-paren), L19H27 (1-paren), L22H10 (1-paren), L23H8 (4-paren), L26H10 (2-paren), L26H13 (1-paren), L27H1 (1-paren), L27H3 (3-paren), L27H18 (1-paren), L28H17 (1-paren), L29H15 (1-paren), L10H0 (1-paren, 2-paren), L10H7 (1-paren, 2-paren), L29H17 (1-paren, 2-paren), L30H22 (2-paren, 3-paren) | L21N13821 (1-paren, 4-paren), L25N2012 (1-paren, 4-paren) |

| Sub-Task | L30H0 (CodeLlama) | L30H16 (GPT-2 XL) | L30H2 (CodeLlama) |
|---|---|---|---|
| One-Paren | #printthestring 8 1 9 print (8 1 9 | #print the string 231 print (231 | #printthestring 8 1 9 print (8 1 9 |
| Two-Paren | #printthestring 691 print(str (691 | #print the string 501 print(str (501 | #printthestring 691 print(str (691 |
| Three-Paren | #printthestring 454 print(str(str (454 | #print the string 245 print(str(str (245 | #printthestring 454 print(str(str (454 |

Table 5: Attention visualization of attention heads. L30H0 of CodeLlama generalizes to all four sub-tasks. L30H16 of GPT-2 XL generalizes to one-, two-, and three-paren. L30H2 of CodeLlama does not generalize to any sub-task.

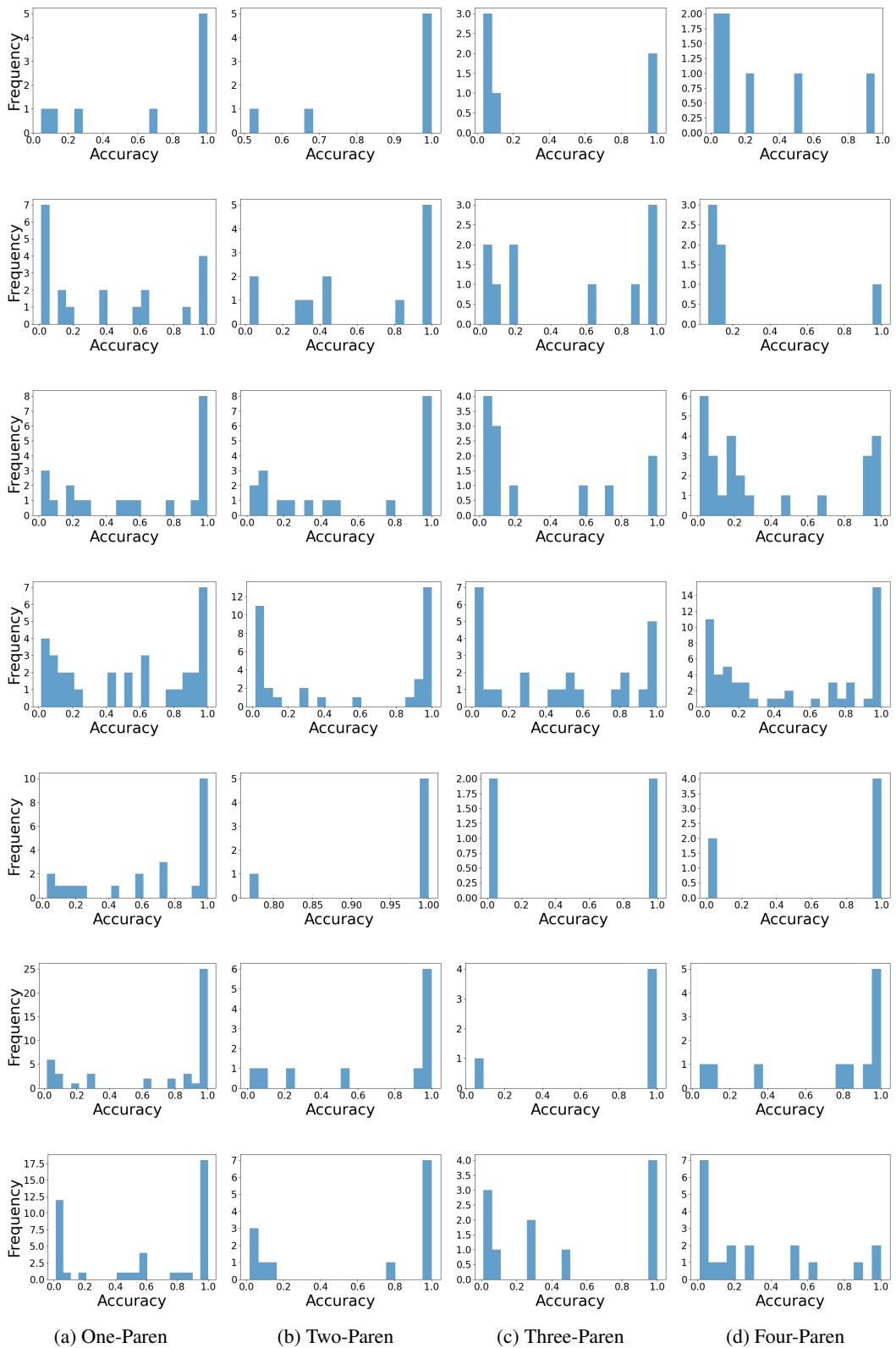

Figure 4: The plots illustrate how attention head accuracy varies across sub-tasks across six models. Each row corresponds to a different model: **top to bottom**—GPT-2 Small, GPT-2 Medium, GPT-2 Large, GPT-2 XL, Llama2-7b, CodeLlama, and Pythia-6.9b. Attention heads with accuracy below 0.01 are excluded to avoid distortion of the distribution due to their high frequency.

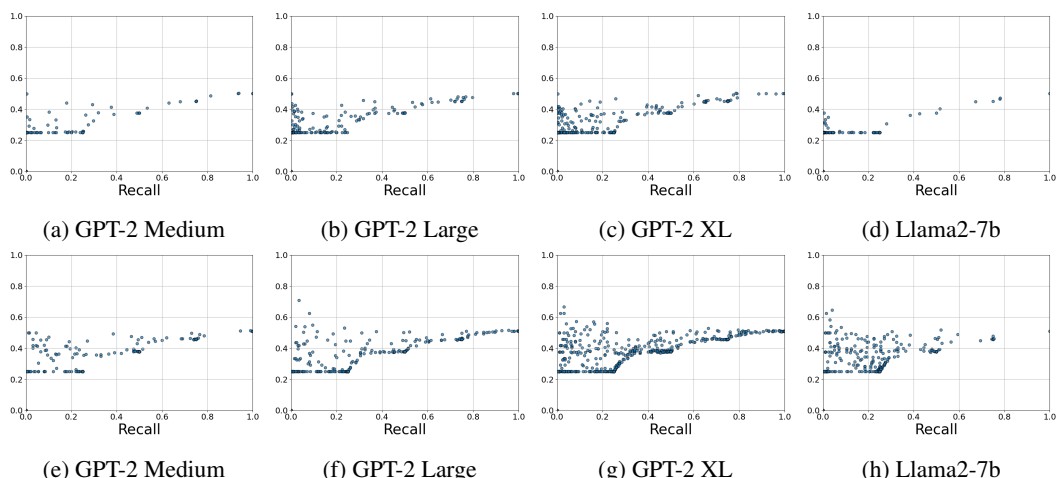

Figure 5: Scatter plots of average precision vs. recall across all sub-tasks for attention heads (**Top**) and FF neurons (**Bottom**)

# C   Details and Results of Circuit Discovery

We follow the same activation patching approach of Nikankin et al. [28] to discover circuits for each sub-task. We only localize important attention heads while retaining all FF layers in the circuit to mitigate the high computational cost of the circuit experiments. In our preliminary exploration, computationally efficient approaches that try to approximate activation patching, such as edge attribution patching [37, 19], did not yield circuits with reasonable faithfulness scores.

**Methodology**   Activation patching requires three forward passes—a clean run, a corrupt run, and a patched run, to evaluate an LM component's effect on the performance of a sub-task, where the predicted next-token of the clean run is the sub-task's correct target label and the corrupt run results in the prediction of a token that isn't the correct target label. In a noising-based approach [33, 20], the patched run takes the input of the clean run, i.e., the clean prompt, as its input and "patches" in the associated cached activation from the corrupted run for an LM component to determine how important that component is for the successful performance of the sub-task. Following Nikankin et al. [28], we define the effect score of an LM component as in Equation 4, where $P_{clean}$ and $P_{patched}$ represent the probability distribution of the next-token predictions for the clean and patched runs, respectively, and $r$ and $r'$ represent the target token of the clean prompt and the counterfactual prompt, respectively. This metric assigns a high effect score to components whose "patched" runs result in a large decrease of the correct token label's probability and/or a large increase of a counterfactual closing parentheses token's probability.

$$E(r, r') = \frac{1}{2}\left[\frac{P_{patched}(r') - P_{clean}(r')}{P_{clean}(r')} + \frac{P_{clean}(r) - P_{patched}(r)}{P_{patched}(r)}\right] \tag{4}$$

**Counterfactual Prompts in Corrupt Runs**   The corruption strategy used for the corrupt runs was resampling ablation [28, 8]. This corruption strategy requires the construction of counterfactual prompts to be utilized in corrupt runs. We constructed these counterfactual prompts for each sub-task by taking its associated clean prompts and increasing its number of open parentheses by one. This was done by replacing a single open-parenthesis token contained in the clean prompt with a token containing two open parentheses for each possible open-parenthesis token position. For example, for the clean prompt under the two-paren sub-task, "#print the string 160\nprint(str(160" → "))", we create its corresponding counterfactual prompt as: "#print the string 160\nprint(str((160 → ")))". For the four-paren sub-task, this token replacement strategy resulted in counterfactual prompts that were unable to be completed within a single-token prediction. In this case, we consider the subsequent token as $r'$. For example, for CodeLlama-7b, the counterfactual prompt for the four-paren sub-task can be "#print the string 615\nprint((str(str(str(615" → "))".

**Dataset Construction**   We utilized the same training set as RASTEER for each sub-task when finding circuits (also described in Section 5.1). Specifically, we collected positive prompts for the sub-task (e.g., prompts demanding ")" as the true token in the one-paren sub-task) as the clean prompts, and additionally filtered out prompts where the model cannot successfully predict the true token. This additional constraint of a clean prompt resulting in a correct target token prediction was imposed to reduce the amount of unintended noise contained in the LM components' effect scores; the same strategy was also adopted by Nikankin et al. [28]. To ensure equal training dataset sizes between RASTEER and the circuit discovery baseline, additional clean prompts that fulfilled this constraint were sampled using the respective sub-task's prompt template, on the condition that the three-digit integer contained in these prompts did not overlap with the three-digit integers contained in the sub-task's test-set prompts. This additional sampling of clean prompts was exclusive to the sub-task's training datasets and was not performed on the sub-task's test datasets to maintain a fair comparison between the methods.

Subsequently, the respective clean prompts were used to generate counterfactual prompts, where we replaced a single open-parenthesis token in the clean prompt with a two-parenthesis token, varying the position of this replacement and optionally replacing the integer number to increase the number of candidate counterfactual prompts. These candidate prompts were then filtered on the constraint that the logit of the clean prompt's target token (e.g., ")" in two-paren) is less than the logit of the corrupt token (e.g., ")))" for corrupt prompts in two-paren) under the counterfactual prompt. The resulting counterfactual prompts for each sub-task were used to form the final dataset for activation

patching. This filtering step was relaxed for model/sub-task combinations, which resulted in an empty filtered counterfactual-prompt set, to allow for a baseline comparison for all model/sub-task combinations where the model could successfully perform the sub-task.

Finally, in order to understand the quality of each discovered circuit, we applied the same dataset construction strategy to sample clean prompts and generate counterfactual prompts for examples on the test set of each sub-task. Note that we created this modified test set only for the purpose of evaluating circuits, whereas in our main evaluation (Section 5.2), all approaches were evaluated on exactly the same test sets.

**Results of Circuit Discovery** We define the *faithfulness* of a circuit as how much of the model's performance on each sub-task, more specifically, the logit value of the clean prompt's correct target token, can be accounted for by the circuit [44, 28]. To evaluate the faithfulness of a prospective circuit, a circuit run, a model run, and a corrupted run are required for every clean and counterfactual prompt pair in the test dataset. For the circuit and model runs, the clean prompt is used as the input, while for the corrupted run, the counterfactual prompt is used as the input. In a circuit run, the output activations of all non-circuit LM components are "patched" with their associated activations from a cached corrupted run. No interventions are required for the model and corrupted runs. Following Nikankin et al. [28], we measure *faithfulness* using the metric in Equation 5, where $NL_{\text{circuit}}(\text{correct})$, $NL_{\text{model}}(\text{correct})$, and $NL_{\text{corrupt}}(\text{correct})$ represent the logit value of the correct target token normalized by the maximal logit. This metric has an upper bound of 1.0 for all model/sub-task combinations and a lower bound of $-1.0$ when the counterfactual prompts for a model/sub-task combination meet the above-mentioned filtering criteria. We consider a circuit faithful if it achieves an average faithfulness score of 0.9 across the test dataset.

$$F(\text{circuit}) = \frac{NL_{\text{circuit}}(\text{correct}) - NL_{\text{corrupt}}(\text{correct})}{NL_{\text{model}}(\text{correct}) - NL_{\text{corrupt}}(\text{correct})} \tag{5}$$

Table 6 presents the faithfulness scores of circuits when only top-K attention heads (sorted by their effect scores) are retained. We only present the result of K when the model achieves a high faithfulness score.

Table 6: Faithfulness scores of circuits found for model/sub-task combinations, where the circuit consists of all FF components and the top-K position-dependent attention heads, in terms of their effect scores. Note that for sub-tasks that a model is unable to achieve non-zero accuracy, activation patching cannot be applied; as a result, we skip finding circuits for them.

| Model | One-Paren | Two-Paren | Three-Paren | Four-Paren |
|---|---|---|---|---|
| GPT-2 Small | 0.96 (K=1) | 0.91 (K=1) | 0.90 (K=1264) | – |
| GPT-2 Medium | 0.98 (K=15) | 0.91 (K=11) | 0.96 (K=3454) | – |
| GPT-2 Large | 0.96 (K=18) | 0.91 (K=4902) | 0.95 (K=6748) | – |
| GPT-2 XL | 0.97 (K=10) | 0.90 (K=8179) | – | – |
| Llama2-7b | 0.92 (K=9) | 0.96 (K=24) | 0.92 (K=50) | – |
| CodeLlama-7b | 0.94 (K=31) | 0.95 (K=23) | 0.97 (K=5) | 0.91 (K=152) |
| Pythia-6.9b | 0.95 (K=6) | 0.90 (K=5) | 0.93 (K=146) | 0.90 (K=157) |

# D Additional Results of RASTEER

## D.1 Results of RASTEER When Steering Both Attention Heads and FF Neurons using F1 Ranking Metric

As shown in Figure 6, steering both attention heads and FF neurons yields better performance than steering either component alone, indicating a complementary effect.

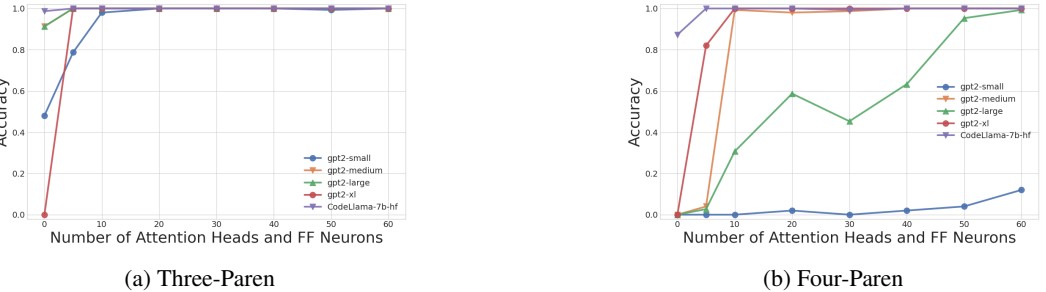

(a) Three-Paren

(b) Four-Paren

Figure 6: Performance of the model after steering both attention heads and FF neurons using RASTEER on the three-paren sub-task (left) and the four-paren sub-task (right).

## D.2 Why Does RASTEER-Attention Heads Fail on the Four-paren for GPT-2 Small?

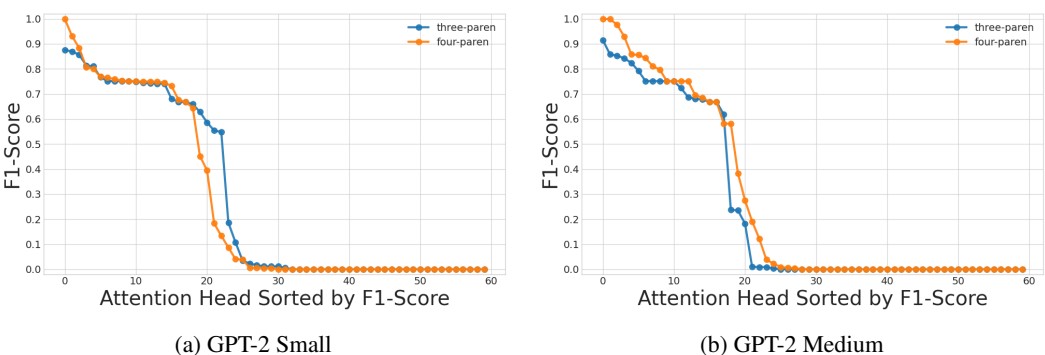

(a) GPT-2 Small

(b) GPT-2 Medium

Figure 7: F1-score distributions of the top-60 RASTEER attention heads sorted by sub-task F1-Score for GPT-2 Small and GPT-2 Medium.

We attempt to answer this question by first examining whether GPT-2 Small lacks attention heads to steer that are accurate for the four-paren sub-task compared to models where RASTEER was successful. In Table 4, we find that while some of the models, i.e., GPT-2 Large, GPT-2 XL, CodeLlama-7b, and Llama2-7b, have a comparatively larger number of attention heads that are accurate on the four-paren sub-task to GPT-2 Small's one four-paren accurate attention head, some models, i.e., GPT-2 Medium and Pythia-6.9b, have a similar or equal number of four-paren accurate attention heads. Thus, it appears that GPT-2 Small's low number of four-paren accurate attention heads is not the sole reason for RASTEER's failure.

Following, we examine if GPT-2 Small has a dramatic weakness in terms of noisy promotion, such that the ground-truth token is under-promoted or other candidate tokens are over-promoted for the four-paren sub-tasks' prompts, compared to models with a similar number of four-paren accurate attention heads. We chose to directly compare the noisy promotion of GPT-2 Small with GPT-2 Medium, instead of Pythia-6.9b, due to these models having the same number of four-paren accurate attention heads and similar performance on the four-paren sub-task before applying RASTEER. We examine the sub-task level F1-scores of the top-60 RASTEER attention heads of each model for the three-paren and four-paren sub-tasks. In Figure 7, we observe similar F1-score distributions for each model/sub-task combination with closely matching counts of attention heads with low/high F1-scores. Indicating that there is a similar number of attention heads across GPT-2 Small and GPT-2 Medium

that will correctly promote the ground-truth token/incorrectly promote other candidate tokens for the four-paren sub-tasks' prompts. Thus, it appears that GPT-2 Small's lack of substantial performance increase after applying RASTEER is not due to a weakness in its noisy promotion, but rather an unexplored factor.

### D.3 Results of RASTEER using Recall and Precision for Component Ranking

As shown in Figure 8, while both RASTEER with precision- or recall-based ranking metrics showed improved performance, the recall-based metric shows slightly better performance on the four-paren sub-task for GPT-2 Large.

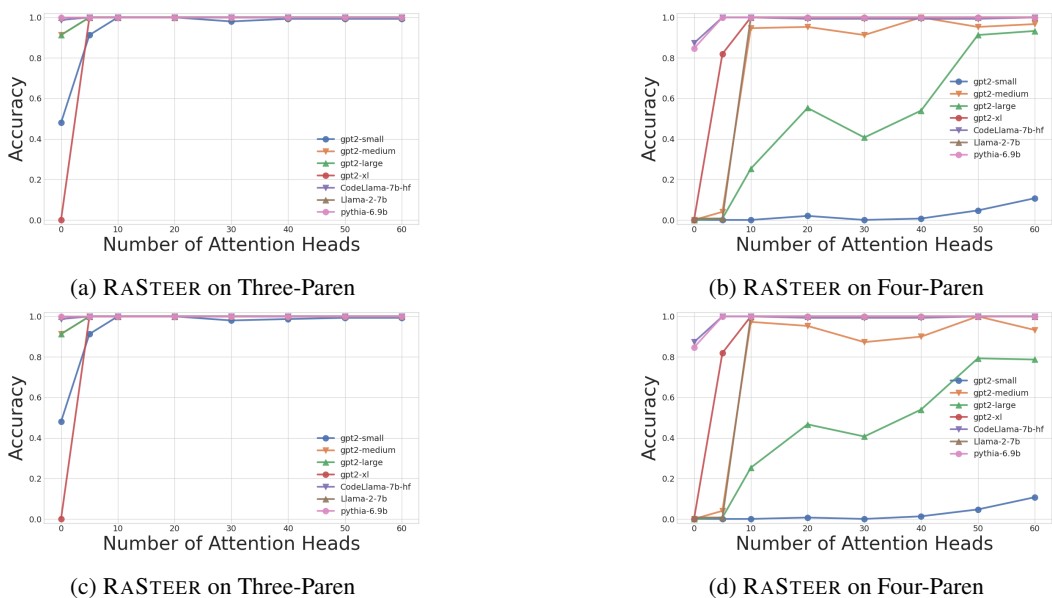

(a) RASTEER on Three-Paren

(b) RASTEER on Four-Paren

(c) RASTEER on Three-Paren

(d) RASTEER on Four-Paren

Figure 8: Comparison of RASTEER performance using recall-based (**top**) and precision-based (**bottom**) ranking metrics when steering attention heads on the three-paren and four-paren sub-tasks.

### D.4 Why Does the Circuit Baseline Underperform RASTEER?

As shown in Figure 3, the circuit baseline performs comparably to RASTEER for larger LMs (over 1B parameters), but fails to show a consistent correlation between steering important attention heads and performance improvement in smaller LMs. To better understand this performance gap, we examine the differences in the attention heads selected by each method—both in settings where the circuit baseline is effective and where it underperforms. We first investigate whether the performance gains in the circuit baseline come from steering the same set of attention heads as RASTEER for the larger LMs. Specifically, we compare the top heads identified by the circuit baseline with the top-$k$ heads identified by RASTEER, with $k$ selected to be the minimal number of heads that raise a model's accuracy to $\sim 100\%$ when steered (referred to as "minimal effective heads" onwards). As shown in Figure 9, the degree of overlap varies between $20 - 100\%$, which shows that the two approaches can identify different sets of important heads. To further isolate the contribution of shared heads on performance improvement, we perform a minimal steering experiment, where for CodeLlama, Pythia, and Llama2, we steer the shared minimal effective heads between the two approaches. For example, for CodeLlama, both approaches need only top-5 heads for steering the model to achieve a 100% accuracy, and we steer the 3 heads overlapped between them (i.e., 0.6 percentage when $x = 5$ in Figure 9). We skip GPT2-XL because of the high overlap. We observe that steering only this minimal shared effective heads led to a similar $\sim$100% accuracy for Llama2 and CodeLlama but no improvement for Pythia. This indicates that the overlapping heads are generally a smaller subset of necessary heads for steering, but not always. This is especially evident in Pythia-6.9b, where steering two disjoint sets of four heads (excluding the shared one) from each method independently achieves $\sim$100% accuracy, highlighting the redundancy of useful mechanisms within the model.

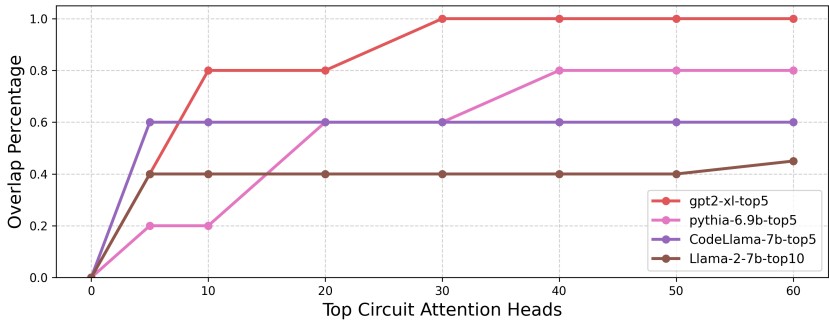

Figure 9: Percentage overlap of the top-$k$ RaSteer attention heads, where $k$=5 for GPT-2 XL, CodeLlama-7b, and Pythia-6.9b and $k$=10 for GPT-2 Medium and Llama2-7b, with the top-60 attention heads from the respective circuit baseline, for models where the steering of a small number of RaSteer attention heads resulted in most of the performance improvement on the four-paren sub-task. For example, the 0.2 overlap percentage for top-10 circuit attention heads indicates 1 shared head with RaSteer's top-5 heads.

Next, we investigate why the circuit baseline is not effective for smaller LMs. Since the circuit analysis consists of heads that will not be discovered by RaSteer and are responsible for intermediate computations such as inter-token information transfer and feature extraction, we posit that these additional heads may introduce instability in steering. To test the effect of steering these additional components, we steer the top-10 RaSteer heads in GPT-2 Medium, which has a performance of $\sim 100\%$, along with 5 circuit-identified heads from non-final token positions (which cannot directly affect the output). This led to a decline in accuracy from $\sim 100\%$ to $4.00\%$, and adding 5 more (10 in total) such heads collapses performance to near $0.00\%$. This indicates that not all task-relevant components are good for steering. Specifically, components that are functionally important for the task but do not directly affect the final logit may introduce instability when promoted.

# E Experiments compute resources

We conduct all experiments on an NVIDIA A100 GPU with 40GB of GPU memory and up to 50GB of CPU memory.

# F Licenses for existing assets

We use the following open-weight models for our experiments.

## F.1 Models

- GPT-2 Models [31] (Modified MIT License at `https://github.com/openai/gpt-2/blob/master/LICENSE`)
- Llama2-7b[40]: Special Llama-2 License at (`https://www.llama.com/license/`)
- Llama-3 8b [17]: Special Llama-3 License at `https://llama.meta.com/llama3/license/`
- CodeLlama-7b [36]: Special Llama-2 License at `https://github.com/meta-llama/llama/blob/main/LICENSE`
- Pythia-6.9b and Pythia-12b [6]: Apache License 2.0 at `https://github.com/EleutherAI/pythia/blob/main/LICENSE`
- Qwen-2.5-3b [38]: Apache License at `https://huggingface.co/Qwen/Qwen2.5-7B/blob/main/LICENSE`

