# OpenReview forum: "Failure by Interference: Language Models Make Balanced Parentheses Errors  When Faulty Mechanisms Overshadow Sound Ones"
_NeurIPS.cc/2025/Conference — NeurIPS 2025 poster_

### Official Review · Reviewer_Wfy8 · 2025-06-28

**Clarity:** 3
**Significance:** 3
**Originality:** 3
**Rating:** 5
**Confidence:** 3

**Summary:**

This paper researches the failures of simple synthetic tasks of LLMs. Authors investigate a parenthesese matching problem, which is simple for humans but hard for LLMs. Authors delve into the mechanism of the phenomenon by examining the computation flow of Transformers. By utilizing an unembedding matrix, the authors can identify neurons that promote correct answers to a certain configurable degree. Authors then discover that LMs develop different mechanisms to solve tasks with different generalization level. For example, an FF neuron at most generalizes to two sub-tasks; the failures are caused by noisy signals. To this end, authors propose sorting components by the degree to which they promote correct responses and scaling the signals of these components accordingly. Results show the proposed method perform better than the circuit baseline.

**Questions:**

* How do authors obtain the unembedding matrix?

**Ethical Concerns:**

["NO or VERY MINOR ethics concerns only"]

**Final Justification:**

The authors address my concerns on the circuit method, the training set (only 350 examples), and clarification questions. Therefore, I maintain my score.

**Limitations:**

yes

**Quality:**

4

**Strengths And Weaknesses:**

### Strengths

* Interesting problem which is valuable to investigate.
* The mechanism research method is sound and clear
* The proposed solution is reasonable

### Weakness

* It is unclear how well the algorithms 1 and 2 make sense because they ignore the long-horizon effect of the neuron. For example, a single neuron may not promote the correct response, but it may be useful in a circuit. Nevertheless, I understand this is hard to quantify.

* The proposed method requires a training dataset, which limits its general application.

Regardless, I believe this is a good analysis paper.

---

> ### Author Rebuttal · Authors · 2025-07-30
>
> We sincerely thank the reviewer for their thoughtful comments and positive assessment of our work. We appreciate the reviewer’s recognition of the importance of the problem, the soundness and clarity of our analysis, and the proposed solution.
>
> **Response to W1:** We have a circuit baseline that should capture components that have these long-horizon indirect effects on the upstream components. However, as seen in Figure 4, when we steer with components found through the circuit analysis, it doesn’t have a strong correlation with the accuracy. This indicates that not all task-relevant components are good for steering. Specifically, components that are functionally important for the task but do not directly affect the final logit may introduce instability when promoted.
>
> **Response to W2:** While our method does require a training set, it is lightweight compared to fine-tuning-based approaches. We use only 350 training examples for the balanced parentheses task, which can be easily synthesized. Additionally, we would also like to note that most steering methods, such as CAA [1], also require training data. That said, we agree with the reviewer that reducing data requirements further is an important direction for future work.
>
> **Response to Q1:** We obtain the unembedding matrix directly from the final layer of the model, which maps the hidden representations of the final layer to vocabulary logits. We will clarify this in the final version of the paper.
>
> **References:**
>
> [1] Rimsky, Nina, et al. "Steering Llama 2 via Contrastive Activation Addition." 2024.

---

> > ### Author Response · Authors · 2025-08-07
> >
> > Dear Reviewer Wfy8,
> >
> > Thank you again for your thoughtful review and positive assessment of our paper. We wanted to briefly check if our rebuttal addressed the weaknesses and questions you raised. As the discussion period ends tomorrow, we’d be grateful if you could confirm whether any concerns remain. Thank you!

---

> > > ### Comment · Reviewer_Wfy8 · 2025-08-08
> > >
> > > Thanks for providing a rebuttal. I decide to maintain my score after reading the rebuttal.

---

### Official Review · Reviewer_K9m9 · 2025-07-02

**Clarity:** 2
**Significance:** 2
**Originality:** 2
**Rating:** 4
**Confidence:** 2

**Summary:**

This study explores why language models (LMs) continue to struggle with fundamental syntactic tasks, such as generating balanced parentheses, despite significant progress in coding capabilities. The research uncovers that LMs employ a diverse array of components—including attention heads and feedforward (FF) neurons—to address sub-tasks, with most components specialized for specific functions and only a few demonstrating broader generalizability. However, these components often exhibit "noisy promotion," simultaneously supporting both correct and incorrect tokens, which results in low precision and recall. To address this, the authors introduce RASTEER, a method that ranks components based on their reliability and amplifies the activations of the most dependable ones. Experiments show that selectively promoting these top-ranked components significantly enhances performance on balanced parentheses tasks, as well as on synthetic datasets, the HumanEval benchmark, and arithmetic reasoning tasks, all while maintaining general coding proficiency.

**Questions:**

[1] Does RASTEER improve model performance on more general benchmarks, as well as on mathematical reasoning benchmarks such as GPQA, MMLU, and GSM8K? The Arithmetic Reasoning tasks presented in this paper are constructed by the authors and involve only two operands, making them relatively simple and limited in scope.
[2] RASTEER selects the reliable components of language models through grouping and sorting, which involves two hyperparameters: a threshold set to 0.7 and a promotion threshold set to 0.5. How did the authors determine these values?
[3] The experiments in this paper are mainly conducted using models such as GPT-2, CodeLlama, and Llama2, which are already outdated and exhibit relatively weak performance. This likely explains the frequent errors observed in tasks like balanced parentheses. Therefore, it remains unclear whether RASTEER would still be effective when applied to more advanced models, such as the Qwen2.5 series.

**Ethical Concerns:**

["NO or VERY MINOR ethics concerns only"]

**Final Justification:**

This paper solves reasoning tasks from a fundamental perspective, which I believe would be contributory in this field.

**Limitations:**

yes

**Quality:**

2

**Strengths And Weaknesses:**

Strengths：
[1] The paper demonstrates a deep understanding of language model (LM) architectures and mechanisms. It provides a detailed analysis of how attention heads and feed-forward (FF) neurons contribute to syntactic tasks like balanced parentheses generation.
[2] The paper is well-structured, progressing logically from problem formulation to methodology, experiments, and conclusions. Key insights (e.g., noisy promotion, mechanism overshadowing) are clearly explained.
Weakness：
[1] The study mainly focuses on synthetic datasets for balanced parentheses, which is very simple and naive. Generalizability to real-world coding scenarios or more complex syntactic structures is not fully explored.
[2] This paper primarily conducts experiments on models such as GPT-2, CodeLlama, and Llama2, which are relatively outdated and have weaker performance. Therefore, it is difficult to determine whether RASTEER would remain effective on more advanced models, such as the Qwen2.5 series.

---

> ### Author Rebuttal · Authors · 2025-07-30
>
> We sincerely thank the reviewer for their thoughtful comments and constructive feedback. We appreciate the reviewer’s recognition of the detailed analysis of how attention heads and feed-forward neurons contribute to syntactic tasks, and the clear, logical structure of the paper. We are also glad that key insights—such as noisy promotion and mechanism overshadowing—were found to be clearly explained.
>
> **Response to W1:** We would like to clarify that the simplified setting was an intentional design choice to enable precise analysis of failure modes in a controlled setting. In more realistic scenarios, errors are often challenging to disentangle, potentially stemming from reasoning failures, knowledge gaps, formatting issues, or task ambiguity. Our controlled setup, by contrast, allowed us to confidently attribute failures to the specific syntactic error under investigation. In addition, synthetic setting also offers a low-cost and tractable way to synthesize dataset, which would be significantly more challenging to annotate in realistic codebases. For these reasons, this approach, where simplified settings distilled from realistic applications is **widely adopted in interpretability research [1, 2, 3, 4]**, and we follow this vein when designing our research. Finally, we also applied RaSTEER to the HumanEval benchmark to check unintended effects, and surprisingly, we find our approach could lead to a small improvement in general coding tasks as well. We believe this provides positive preliminary results for the broader applicability of our approach beyond this initial setting.
>
> **Response (W2 and Q3):** We appreciate the reviewer’s concern and agree that evaluating on more recent models is important. To address this, we conducted additional experiments on **Qwen2.5-3B**, as suggested by the reviewer. We found that Qwen2.5-3B has 100% accuracy in the one-paren and two-paren subtasks. While for the more challenging three-paren and four-paren subtasks, the model achieved 92.66% and 46.66% accuracy, respectively. Next, we applied RaSTEER by promoting the top 20 attention heads, which improved the accuracy on the three-paren subtask from 92.66% to 98.00%, and on the four-paren subtask from 46.66% to 77.33%. We observed no further gains when promoting additional heads. We also verified that the RaSTEER did not harm general coding performance on HumanEval; instead accuracy improved the accuracy slightly from 33.53% to 35.36%. Additionally, we have also applied RaSTEER to **Llama3-8B** (misnamed as GPT-3 8B in the current version; we will correct this in the final version), a relatively new model, for the arithmetic task in Section 5.4 and observed improvement (up to 6.00%) in accuracy. These results suggest that RaSTEER is effective when applied to more recent and advanced models such as Qwen2.5 and Llama3 series.
>
> **Response to Q1:** In this study, we focused on implicit reasoning tasks, where the model is required to produce the final answer in a single decoding step without generating intermediate reasoning. In contrast, benchmarks such as MMLU and GSM8K involve explicit chain-of-thought (CoT) reasoning, which requires multi-step generation and was considered outside the scope of this work. We believe that understanding the implicit reasoning of the model is a critical step toward interpreting and improving more complex explicit reasoning behaviors. Following common practice in prior work [1, 2, 3], we used a synthetic dataset to study the mechanisms involved in implicit reasoning. That said, we agree with the reviewer and acknowledge that evaluating RaSTEER on more challenging benchmarks like MMLU, GSM8K, and GPQA is an important direction for future work.
>
> **Response to Q2:** The **accuracy threshold** of 0.7 for grouping components was selected based on empirical observations of accuracy distributions across sub-tasks (Figures 1 and 5). We found that this value effectively separates components with high accuracy in at least one sub-task from others.  For the **promotion threshold**, we initially used 0.5 as a reasonable threshold for the analysis in Section 3, then validated it later through tuning on the development set for RaSTEER. Specifically, we evaluated thresholds in the range [0.4,0.5,0.6,0.7,0.8,0.9] and found that 0.5 consistently achieved strong performance. For example, on the four-paren sub-task with GPT-2 Small (top-60 heads), 0.5 yielded 16.8% accuracy—outperforming other thresholds, which ranged between 5.3% and 14.0%. Notably, we found that the promotion threshold had less impact on larger models (≥1B parameters), where promoting just a small number of reliable components is often sufficient for performance gains. In summary, we posit that the promotion threshold balances between false positives and false negatives: thresholds below 0.5 risk including too many components that do not meaningfully promote the target token, while thresholds above 0.5 risk excluding components that are genuinely helpful. We will include these empirical details in the updated version of the paper.
>
> **References:**
>
> [1] Yaniv Nikankin, Anja Reusch, Aaron Mueller, and Yonatan Belinkov. Arithmetic without algorithms: Language models solve math with a bag of heuristics. In The Thirteenth International Conference on Learning Representations, 2025.
>
> [2] Stolfo, Alessandro, Yonatan Belinkov, and Mrinmaya Sachan. "A Mechanistic Interpretation of Arithmetic Reasoning in Language Models using Causal Mediation Analysis." Proceedings of the 2023 Conference on Empirical Methods in Natural Language Processing. 2023.
>
> [3] Nanda, Neel, et al. "Progress measures for grokking via mechanistic interpretability." The Eleventh International Conference on Learning Representations. 2023.
>
> [4] Wang, Kevin Ro, et al. "Interpretability in the Wild: a Circuit for Indirect Object Identification in GPT-2 Small." The Eleventh International Conference on Learning Representations. 2023.

---

> > ### Comment · Reviewer_K9m9 · 2025-08-06
> >
> > Thank the authors for their detailed response. My concerns have been addressed. I will increase my original rating to 4.

---

> > > ### Author Response · Authors · 2025-08-06
> > >
> > > Thank you for taking the time to review our rebuttal. We're glad to hear that the weaknesses and questions have been addressed. We will incorporate the clarifications into our final manuscript.
> > >
> > > We appreciate your increase in the overall score. If you feel the improvements warrant it, may we request you to update the sub-category scores (quality, clarity, significance, and originality) accordingly, to make the evaluation more consistent? Thanks!

---

### Official Review · Reviewer_kKr8 · 2025-07-02

**Clarity:** 3
**Significance:** 3
**Originality:** 3
**Rating:** 5
**Confidence:** 4

**Summary:**

The paper investigates the difficulty in generating balanced parantheses by different LLMs. The authors identify the LLM components those are responsible for introducing inaccuracies in balanced parantheses generation. In order to mitigate this, the paper proposes a steering method called RASTEER to rank the components based on their reliability and contribution in final logits computation. The authors report substantial improvements in the LLMs for the task and also demonstrate the overall efficacy of their approach in mathematical reasoning.

**Questions:**

1. The authors should mention the selection criteria of the LLMs.
2. It is not clear why larger LLMs (> 7B parameters) are not explored.
3. The authors can add a detailed section on different prompts describing each undertaken strategy in the appendix.

**Ethical Concerns:**

["NO or VERY MINOR ethics concerns only"]

**Final Justification:**

The authors have given satisfactory responses to the queries regarding the prompting strategies and the experiments with models > 7B parameters.

**Limitations:**

Yes

**Quality:**

3

**Strengths And Weaknesses:**

The authors analyze components such as attention heads, feed forward neural networks that contribute majorly in the final logit computations. They also propose algorithms to identify the correct tokens over distractors and promote correct tokens with predefined thresholds. When the NLP community is trying out various complex tasks on LLMs, the authors attempt to unearth the reason behind errors encountered for a simple task of balancing parantheses.

The authors should mention the selection criteria of the LLMs. It is not clear why larger LLMs (> 7B parameters) are not explored. Different prompts detailing the strategies would have helped to understand the paper better.

---

> ### Author Rebuttal · Authors · 2025-07-30
>
> We sincerely thank the reviewer for their thoughtful comments and positive assessment of our work. We appreciate the reviewer’s recognition of our proposed algorithms to identify components that promote the correct tokens over distractors.
>
> **Response to W1, Q1, and Q2:** To demonstrate that insights obtained from our study and proposed approach generalize across various model families and sizes, we considered the models that varied in the following aspects.
>
> - Model Size (117M to 7b models): We only considered model sizes below 7b because of computational budget constraints. We would also like to mention that many prior interpretability studies [1, 2, 3, 4] also consider models of similar sizes for similar studies.
>
> - Model families:
>
>    - Base models (GPT-2, Llama2-7b, Pythia)
>
>     - Code-specific LM (CodeLlama-7b)
>
> - Performance diversity: We selected models with different accuracy on our synthetic dataset (0% to 100%) and on HumanEval (0% to 30.48%).
>
> Although we face computational constraints, we agree that extending the analysis to larger models is an important direction for future work and will clarify this limitation in the revised version. To this end, we conducted **additional experiments on Pythia-12b for the arithmetic task**. RaSTEER improved accuracy across operations—addition, subtraction, multiplication, and division—from 17.33%, 6.00%, 8.00%, 27.00% to 24.66%, 7.33%, 21.00%, 36.33%, respectively. These results are consistent with improvements seen in smaller models and demonstrate RaSTEER’s potential applicability to larger-scale LMs. We will include the selection criteria of the LLMs and new results from Pythia-12b in the final version of the paper.
>
> **Response to w2 and Q3:** Thank you for the suggestion. We describe our prompt design strategy for the balanced parentheses task in Section 2.2, and provide a brief discussion of the arithmetic reasoning prompt in Section 5.4. For HumanEval, we use the original dataset-provided questions without additional prompting. We will include a more detailed description of all prompt strategies in the Appendix of the final version to improve clarity and reproducibility.
>
> **References:**
>
> [1] Yaniv Nikankin, Anja Reusch, Aaron Mueller, and Yonatan Belinkov. Arithmetic without algorithms: Language models solve math with a bag of heuristics. In The Thirteenth International Conference on Learning Representations, 2025.
>
> [2] Stolfo, Alessandro, Yonatan Belinkov, and Mrinmaya Sachan. "A Mechanistic Interpretation of Arithmetic Reasoning in Language Models using Causal Mediation Analysis." Proceedings of the 2023 Conference on Empirical Methods in Natural Language Processing. 2023.
>
> [3] Nanda, Neel, et al. "Progress measures for grokking via mechanistic interpretability." The Eleventh International Conference on Learning Representations. 2023.
>
> [4] Wang, Kevin Ro, et al. "Interpretability in the Wild: a Circuit for Indirect Object Identification in GPT-2 Small." The Eleventh International Conference on Learning Representations. 2023.

---

> > ### Comment · Reviewer_kKr8 · 2025-08-05
> > **Response to Author Rebuttal**
> >
> > Thanks for response. The responses to my queries will the final manuscript a solid contribution.

---

> > > ### Author Response · Authors · 2025-08-06
> > >
> > > Thank you for taking the time to review our rebuttal. We're glad to hear that the weaknesses and questions have been addressed. We will include the suggestions and results from the new experiment in the final manuscript.

---

### Official Review · Reviewer_6Wg5 · 2025-07-03

**Clarity:** 3
**Significance:** 2
**Originality:** 2
**Rating:** 4
**Confidence:** 3

**Summary:**

The stated goal of this submission is to analyze the internal mechanisms used by LMs to generate balanced parentheses. Specifically, the submission makes the assumption that all opened parentheses should be closed by generating a single token. In practice, this means that the LM is prompted with code having 1 to 4 open parentheses and now should generate exactly one of the following tokens: ")", "))", ")))", or "))))".

The analysis of internal mechanisms consists in ranking what the authors term "components", i.e., individual attention heads and individual neurons in feed-forward layers. The authors rank components according to wether they promote or suppress the correct output token. Concretely, components are scored via the LogitLens, i.e., each component's output is projected onto the vocabulary using the unembedding matrix, resulting in a logit score which is taken to quantify the component's contribution to the model generating the output token or not. By comparing if a component's logits of the correct token are larger than the logits all wrong token choices, the authors also compute the task accuracy of that component.

Analyzing these scores, the authors find that some components promote the correct output tokens while others promote wrong ones. The former are termed "sound mechanisms" and the latter "faulty mechanisms".
Thresholding on the scores, the authors identify task-relevant subgraphs of the entire computation graph, i.e., sets consisting of attention heads and FF neurons. The authors show that scaling the activations of all task-relevant subgraphs by a factor >1 improves task parentheses balancing without adverse effects on general code generation capabilities.

In addition to parentheses balancing, the authors also apply this method to a basic arithmetic task in the range of 0-999 and show that activation scaling of the identified components improves performance.

**Questions:**

Q1. Regarding weakness W1 above, my concern is that since, according to my current understanding, the proposed method increases the output probabilities of all valid tokens and decreases the output probabilities of all non-valid tokens. So a simple baseline would be to restrict the original models to generating only the four closing parentheses tokens in the parentheses balancing task and to generating only integer tokens in the arithmetic task. How would the proposed method compare to this baseline? (or if this

Q2. Regarding W2, One counter-argument could be that one should post-process LM output with a linter or that all code in HumanEval is somehow processed into a standard format (e.g., removing line breaks where possible, thereby increasing the number of adjacent parentheses). I'm happy to remove this weakness if this done.

**Ethical Concerns:**

["NO or VERY MINOR ethics concerns only"]

**Final Justification:**

While the author response addressed two of the three weaknesses I identified (W1, W3), one weakness (W2, limited applicability) remains and prevents me from giving a higher score.

**Limitations:**

yes

**Quality:**

3

**Strengths And Weaknesses:**

Strengths:

S1. The paper successfully localizes a small set of components that are causally relevant for the task of interest on two tasks across several models, suggesting that the proposed method is generally useful

S2. The paper is written well and is easy to follow

Weaknesses:

W1. The analysis and identification of components is limited to promotion and suppression of output tokens. It is unclear if the identified components are specific to the task of interest, or if they are simply general biases that always increase or decrease the output token probabilities regardless of context. For example, if one were to apply the logitlens to a randomly-initialized model and rank attention heads and FF neurons according to their logit contributions to specific tokens, one would also find "sound mechanisms" that promote these tokens and "faulty mechanisms" that suppress target tokens and/or promote non-target tokens.

W2. The proposed method is only applicable to tasks in which the desired output is a single token from a small set of tokens that are known in advance

W3. Related to W2, the balanced parentheses task is analyzed only in a very specific setting with 1 to 4 parentheses to be closed in a single step on a single line. My intuition is that a lot of naturally occurring code would split closing parentheses across multiple lines with various levels of indentation, making the considered setting somewhat artificial. This might be relevant for the author's argument that the interventions on the identified components do not affect general coding capabilities, since the prevalence of single-token closed parentheses might be relatively small in the evaluation data, and hence even randomly replacing ")))" or "))))" with another set of closing parenthesis might not lead to a large performance drop, especially since the benchmark (HumanEval) is evaluating Python code, which generally does not have too many parentheses.

---

> ### Author Rebuttal · Authors · 2025-07-30
>
> We sincerely thank the reviewer for their thoughtful comments and constructive feedback. We appreciate the reviewer’s recognition that the study was able to localize causally relevant components across multiple tasks and models, as well as their feedback on the clarity and readability of our writing.
>
> **Response to W1 and Q1:**
>
> **1. Our method selects task-specific and context-sensitive components, not components with general biases to a set of valid tokens.**
>
> - **Clarification:** First, we would like to clarify a potential misunderstanding. Our approach does not work by “increasing the output probabilities of all valid tokens (e.g., parentheses tokens) and decreasing the output probabilities of all non-valid tokens”. Instead, our approach works by promoting the output of LM components that can increase the output probabilities of the correct valid token out of all valid ones. This is achieved by analyzing the task-specific performance of each LM component at the prediction position (Section 3).
>
> - **Metric Design Reinforces Precision:** Our component ranking uses F1-score, which penalizes components that promote correct tokens in the wrong contexts. Components with general bias would have poor precision and hence receive low scores. Consequently, our approach is unlikely to promote components that are simply general biases to a set of tokens.
>
> - **Evidence from Subtask Accuracy:** In Section 3.2, we discover several components with high accuracy across multiple subtasks. For instance, attention head L30H0 consistently achieved nearly 100% accuracy across all subtasks with different inputs. This indicates a clear context-dependent behavior of discovered components.
>
> - **Shared Component Set Across Subtasks in Steering:** We do not promote different sets of components for different subtasks (see Line 234); instead, we steer the same set of components across all subtasks. This further reduces the risk of selecting components that only help with a specific subtask and instead ensures broader, context-sensitive behavior.
>
> - **Hypothetical randomly-initialized model:** We agree that a randomly initialized model may consist of components that exhibit general biases. However, it is unlikely to contain components with the high, context-sensitive accuracy we observe (e.g., L30H0). We will clarify this distinction in the final version.
>
>
>
> **2. Would a restricted-output baseline (valid tokens only) match RaSTEER?**
>
> We believe the suggested baseline is equivalent to comparing with the full model because models rarely produce invalid tokens for the task input. In our study, we found **all models already generate valid tokens even when they make mistakes**, with only 1 exception in GPT-2 Small and 3 in CodeLlama-7b. This suggests the model understands the task (producing balanced parentheses) and is **already biased towards producing valid tokens**, but struggles with selecting the correct one among the valid tokens. Under this situation, the suggested baseline would not lead to different results.
>
> **Response to W2:**
>
> While we primarily focused on the balanced parentheses task that consisted of only 4 valid output tokens, we also apply our approach to a two-operand arithmetic task (Section 5.4) where the set of valid target tokens includes all the integers from [0, 1000] that tokenizes to single tokens. Despite this significantly larger set of valid tokens, our method still achieved meaningful performance gains—up to 20.25% improvement in Pythia-6.9b—suggesting that RaSTEER is effective even when the number of valid output tokens is large. In addition, as the reviewer pointed out, our experiment on HumanEval further shows that our approach works well when there is a large set of valid tokens.
>
> **Response to Q2:** We may have misunderstood the reviewer’s suggestion because of an incomplete sentence. We would like to clarify that when we applied our approach to the HumanEval benchmark, we did not use any syntax checker; we steered the model over all decoding steps because when a model requires to generate closing parentheses cannot be trivially "known in advance". However, if the suggestion was to use Linter to detect syntactic errors and apply our method only when needed, we agree that this is a reasonable and practical integration. Currently, our approach is designed to proactively steer model behavior during the entire generation, rather than relying on post-hoc correction. However, we acknowledge that when the downstream effects of steering on overall code generation task are uncertain or untested, it may be preferable to apply conditionally—e.g., only when a linter detects failure.
>
> **Response to W3:** We acknowledge that our synthetic dataset focuses on a simplified setting with 1–4 closing parentheses on a single line, which may not capture the full variety of naturally occurring code. To assess how artificial this setup is, we analyzed the generation of Llama2-7b, CodeLlama-7b, and GPT-2-XL in HumanEval test set. We did not find any instances where closing parentheses were split across multiple lines. While this does not negate the reviewer’s broader point—especially for other programming languages or codebases with different formatting conventions—it suggests that **our setting does occur naturally** and is especially common in Python, where multi-line closing parentheses are less frequent.
>
> In regard to the concern that RaSTEER may have little impact on general coding performance due to the infrequency of closing parentheses, we analyzed HumanEval generation from Llama2-7b and found that the model is tasked with producing one of the closing parentheses tokens in at least **69.63%** of the generation steps needed to produce a valid line of code. This indicates that parentheses tokens are indeed frequent for HumanEval benchmark. In addition, we would like to clarify that RaSTEER is applied at every generation step, not just when predicting closing parentheses (because our goal is to steer the general model behavior in deciding balanced parentheses, and we cannot foresee when a model will predict parentheses). Therefore, the fact that RaSTEER does not hurt a model's generic coding performance even when they are activated for every generation step indicates that it selectively encourages better syntactic completion without degrading semantic fluency.

---

> ### Comment · Reviewer_6Wg5 · 2025-08-06
> **concerns addressed, will raise scores accordingly**
>
> Thank you for your comprehensive response. The response clears up a misunderstanding on my part (W1) and addresses weakness W3, as well as my questions.

---

> > ### Author Response · Authors · 2025-08-06
> >
> > Thank you for taking the time to review our rebuttal. We're glad to hear that the clarifications resolved W1 and W3, as well as the questions. We wanted to check **whether W2 has been addressed**, as it wasn’t explicitly mentioned in the comment. Please let us know if there’s anything further we can clarify.

---

### Note · Authors · 2025-08-11

We thank the ACs for handling our submission and the 4 reviewers for their thorough feedback and **positive assessments** during the review and discussion phases. To the best of our understanding, there are **no outstanding concerns** from any reviewer.

**Reviewer 6Wg5** highlighted that our proposed approach is **generally useful** and it successfully localizes a small set of components causally responsible for model performance for the task of interest, and found the paper well-written and easy to follow.

- Post discussion: The reviewer confirmed that the weaknesses they described in the initial review (i.e., confusion about whether our approach identifies task-general LM components and its application to general code generation tasks), along with their questions, were addressed.

**Reviewer kKr8** also highlighted the proposed approach as one of the strengths of the paper and praised our focus on uncovering the reasons behind model errors.

- Post discussion: The reviewer confirmed that our responses to their queries (i.e., selection criteria of LMs in experiments, and generalization of our approach to a larger LM) “will (make) the final manuscript a solid contribution”.

**Reviewer K9m9** commended our detailed analysis of LM components for balanced parentheses generation, noting the paper’s clear organization and writing, and also appreciated insights such as noisy promotion and the overshadowing of reliable mechanisms by noisy ones.

- Post discussion: The reviewer confirmed that their “concerns have been addressed” and that they will increase their rating. We addressed their concern about the use of synthetic dataset and the need to experiment with the latest LM.

**Reviewer Wfy8** appreciated that our paper tackles “an interesting problem which is valuable to investigate”, the soundness of our research method, and the proposed approach.

- Post discussion: The reviewer decided to maintain their (positive) initial score.

In addition, we also added two new sets of experiments:
1. Reviewer kKr8’s concern about missing >7B parameter experiments was addressed by adding Pythia-12B results for the arithmetic task, showing improvements consistent with smaller models.
2. Reviewer K9m9’s concern about the applicability of our approach to newer LMs was addressed by testing Qwen2.5-3B, which resulted in improvements of up to 33.67%.

We hope this final remark provides a good summary of our reviews and the rebuttal outcomes. Thank you!

---

### Decision · Program_Chairs · 2025-09-17

**Decision:**

Accept (poster)

**Comment:**

The paper does a mechanistic analysis of the failure of LMs of varying sizes (124M–7B) on the simple task of generating balanced parentheses. The authors identify several components of the trained LMs that make their own predictions for the completion task, where some are sound and some are faulty. They argue that errors occur when there are more faulty vs sound mechanisms and to address this propose a steering mechanism that improves performance drastically on this task. They show some gains from this idea on a more general arithmetic reasoning task (still idealized).

The paper steers away from doing a full mechanistic analysis but rather identifying a set of components that affect performance, and boost the correct ones to improve performance. This to my knowledge is a novel steering approach beyond directional steering. While this is computationally very expensive, requires a training set, and has been evaluated only for simple tasks with a single output token answer, I (and the reviewers) believe that it is a useful contribution to the literature. Additionally, during the rebuttal phase, the authors have provided additional experiments to alleviate most of the concerns of the reviewers.

I encourage the authors to add the additional proposed experiments to the camera-ready and a discussion about how to extend beyond single token answers, and the impact of their steering approach on the general performance of the models.